# Asymmetric learning and adaptability to changes in relational structure during transitive inference
Thomas A. Graham [1,2,3,4] ✉ & Bernhard Spitzer[3,5]

Humans and other animals can generalise from local to global relationships in a transitive manner. Recent research has shown that asymmetrically biased learning, where beliefs about only the winners (or losers) of local comparisons are updated, is well-suited for inferring relational structures from sparse feedback. However, less is known about how belief-updating biases intersect with humans' capacity to adapt to changes in relational structure, where re-valuing an item may have downstream implications for inferential knowledge pertaining to unchanged items. We designed a transitive inference paradigm involving one of two possible changepoints for which an asymmetric (winner- or loser-biased) learning rule was more or less optimal. Participants (N = 83) exhibited differential sensitivity to changes in relational structure: whereas participants readily learned that a hitherto low-ranking item increased its rank ('up' condition), moving a high-ranking item down the hierarchy impaired downstream inferential knowledge ('down' condition). Behaviour was best captured by a reinforcement learning model which exhibited an initially winner-biased learning strategy that was nonetheless adaptable – that is, while this winner bias predominantly limited participants' flexibility in the 'down' condition, well-performing participants were able to reduce or even reverse their winner bias in order to appropriately accommodate the relational change. Our results indicate that asymmetric learning not only accounts for efficient inference of latent relational structures but also for differences in the ease with which learners accommodate structural changes.

Humans readily learn how items rank on a variety of latent scales, such as those pertaining to hedonic or economic value, or social influence. Such representations of rank permit novel inferences of indirectly related states or entities. For instance, knowing that A < B and B < C enables one to infer, through transitive inference (TI), that A < C. TI has been widely studied in humans, non-human primates, rats and birds alike[1–4]. Under TI learning regimes, training trials offer participants trial-and-error feedback about pairwise comparisons between items of neighbouring rank, which must then be used to infer unseen test relations between non-neighbouring items. In requiring agents to use the outcomes of pairwise comparisons to update their estimates of the rankings within a linear set, TI paradigms lend themselves to the application of simple reinforcement learning (RL) algorithms that model the influence of choice feedback on the subjective value of the compared items. Recent work adopting this approach demonstrated that TI learning is characterised by, and indeed benefits from, an asymmetric learning rule under which either the winner (or the loser) of a pair is preferentially updated[5]. Specifically, this benefit emerged in simple RL models furnished with separable, or 'asymmetric', learning rates for updating winners and losers, with most participants displaying a bias towards updating winners. This cognitive distortion during inferential learning fits into a wider body of literature on human biases towards positive[6,7] or confirmatory feedback signals[8–11].

The constantly changing nature of an agent's environment necessitates that any capacity for relational learning must exhibit adaptability, while also ensuring robustness[12]. The learning dynamics underlying humans' ability to adapt to volatile reward environments have been studied in tasks involving changepoints or reversals[13–15]. Likewise, sensory preconditioning paradigms have been used to investigate the conditions under which relational representations are retrospectively re-evaluated via relearning associations between rewarded and indirectly related stimuli, or through inference at the

[1]Max Planck Institute for Biological Cybernetics, Tübingen, Germany. [2]Max Planck Institute for Human Cognitive and Brain Sciences, Leipzig, Germany. [3]Research Group Adaptive Memory and Decision Making, Max Planck Institute for Human Development, Berlin, Germany. [4]Max Planck School of Cognition, Leipzig, Germany. [5]Chair of Biopsychology, Faculty of Psychology, TUD Dresden University of Technology, Dresden, Germany. ✉e-mail: thomas.graham@tuebingen.mpg.de

time of choice[16,17]. These studies have demonstrated humans' ability to infer how changes in local reward feedback pertain to indirectly related stimuli, underscoring the utility of changepoint manipulations in probing inferential learning capabilities.

Studying changepoints in larger relational structures allows one to investigate how agents rapidly modify existing knowledge in response to minimal new information[2,18]. Less is known, however, about how such 'few-shot' local relational changes impact downstream inferential knowledge, nor how this capacity to adapt to changes in relational structure intersects with well-documented belief-updating biases in humans. Consider a sports league where a spectator learns how the teams rank with respect to one another based on the outcomes of head-to-head matches between them. Halfway through the season, the unexpected loss of the reigning champions against a team sitting at the bottom of the hierarchy may be indicative of the former's fall from grace, and/or the latter's resurgence. Ascertaining which team's

ranking has changed will thus determine how much one needs to update one's predictions about how this team will fare against others in the league, while ensuring minimal disruption to knowledge pertaining to the relations between teams whose performance remains unchanged (Fig. 1A). Interestingly, a corollary of the asymmetric RL framework is that the ease with which such changes in relational structure are accommodated, and thus any resultant impact on downstream inferential knowledge, should vary as a function of the asymmetry in the agent's belief updates (see Figs. 1C and S1 for simulations). If humans are biased towards preferentially increasing their estimates of winners, then the sudden decline of the hitherto best team to the bottom of the leaderboard should be less readily accommodated than the rapid ascendency of the worst team to the top of the table. Importantly, the former scenario should also induce more disruption to knowledge of intermediate relations in the table, as the agent's readiness to (incorrectly) increase their estimate of the quality of the low-ranking team effectively

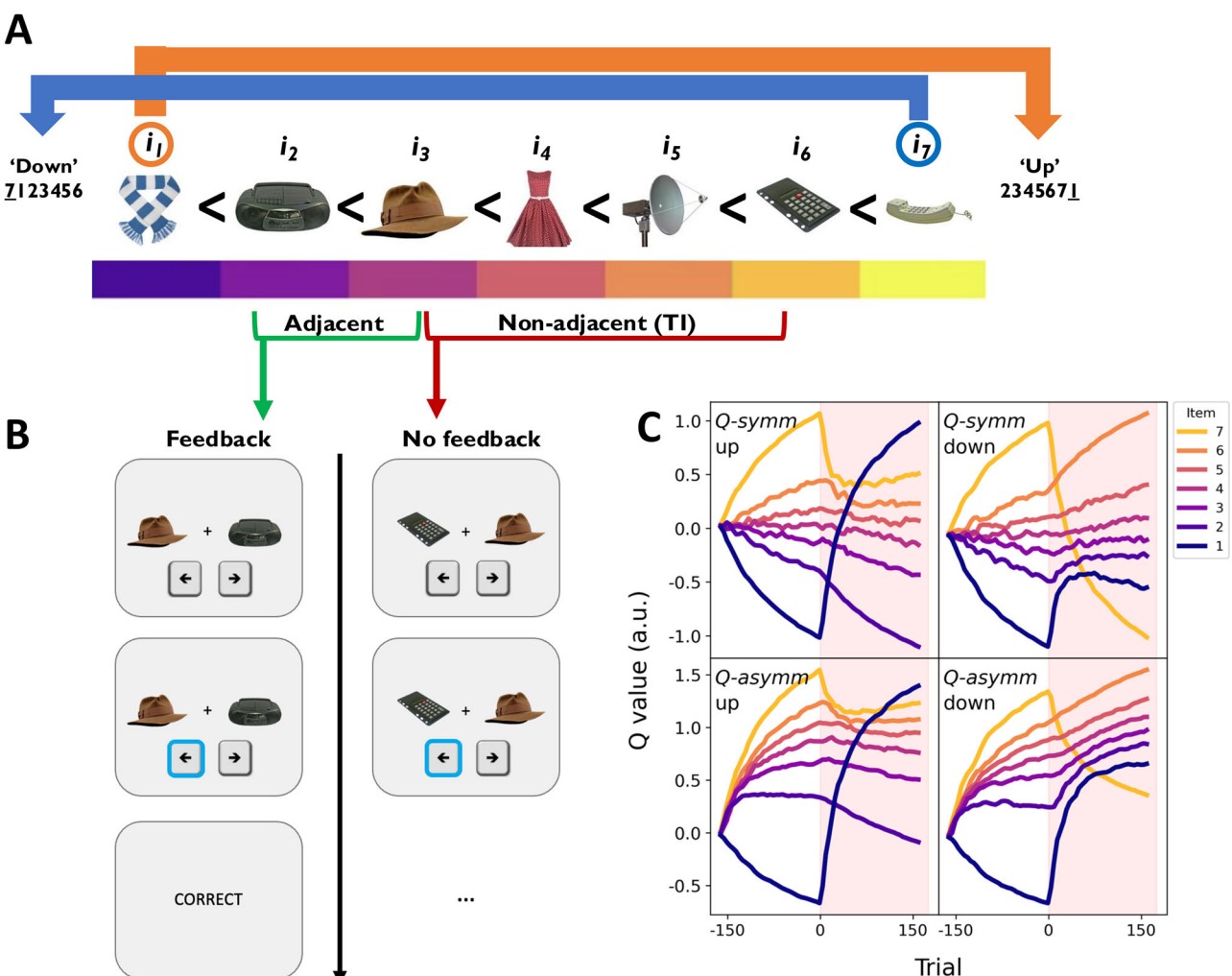

**Fig. 1 | Experimental paradigm and model simulations. A** Example 'cnarciness' rankings of a set of seven items in an ordinal hierarchy. After three blocks, the ground truth structure changed in one of two possible ways: in the 'down' group (blue), the most cnarcy item $i_7$ (here, the telephone) moved to the bottom of the hierarchy, whereas in the 'up' group (orange), the least cnarcy item $i_1$ (here, the scarf) moved to the top of the hierarchy. **B** On each trial, participants were asked to choose which of two items they believed to be the most cnarcy. Binary feedback was delivered on adjacent trials containing items neighbouring in rank (green), while TI comparisons between non-adjacent items offered no feedback (red). **C** Simulated item value estimates ('Q values') under the symmetric agent Q-symm (top row) and the asymmetric agent Q-asymm (bottom row) for the 'up' and 'down' experimental conditions (left and right columns, respectively). X-axis refers to the trial relative to

the first post-changepoint trial, where the red shaded half of each panel represents the post-changepoint phase of the experiment. Whereas non-anchor item value estimates are equally discriminable following both changepoints under Q-symm's symmetric learning rule, Q-asymm predicts impaired discriminability of item values in the 'down' condition relative to the 'up' condition. Specifically, after an initial positive surge in the 'down' agent's value estimate of $Q(i_1)$, Q-asymm's inflexibility to reduce the value of any given comparison's loser results in further compression of the entire value space, since each item needs to further move 'up' the hierarchy in order to correctly accommodate the decline in $i_7$'s value that the agent lacks the sensitivity to incorporate efficiently (cf. Fig. S1). Models were simulated using parameter ranges consistent with participant learning asymmetries reported by Ciranka et al.[5].

'propagates' up the rest of the hierarchy, impairing one's ability to make clear predictions about the relative strengths of these mid-table teams. The relative difficulty with which this former change in ground truth structure is learned would also, in turn, reduce the discriminability of mid-table teams whose rankings remain unchanged, and thus disrupt the agent's inferential knowledge with respect to the middle of the transitive hierarchy.

Accordingly, there is evidence to suggest that the preferential integration of positive reward prediction errors can lead to choice inertia when the best and worst options in a two-armed bandit are flipped[19–21]. Likewise, humans are more reluctant to revise their subjective beliefs about the quality of a deteriorating foraging environment, relative to an environment whose reward rate improves[22]. While these studies support the idea that positively biased agents are more sensitive to positive changes in the values of options and reward environments, the prediction that the biased reorganisation of relational knowledge should have a downstream impact on unchanged elements of a transitive hierarchy remains untested. Moreover, while these predictions are made under the assumption of a static degree of learning asymmetry, introducing a changepoint in a TI learning paradigm also allows one to explore whether learning asymmetries may dynamically adjust or even reverse in a task-dependent manner, a possibility for which empirical evidence in other learning regimes is mixed[23–25].

Here, we therefore sought to investigate whether biased learning policies confer different levels of (in-)flexibility to changes in an environment's relational structure. Participants ($N = 83$) performed a TI paradigm involving one of two possible changepoints for which a winner-biased learning rule was more or less optimal. In addition to replicating previously observed learning asymmetries in the pre-changepoint task phase, we found evidence supporting our model prediction that such biased learning strategies differentially advantage agents' ability to accommodate directional shifts in the environment's underlying relational structure. Computational modelling of behaviour further revealed that such differential sensitivity was best captured by an extension of our asymmetric RL model whose degree of learning rate asymmetry was itself responsive to relational changes. We therefore provide evidence that humans' winner-biased learning asymmetries shape their capacity to update their relational knowledge, unifying our present findings with previous research into belief-updating biases.

## Methods

### Stimuli, Task and Procedure

The behavioural task was an adapted version of the TI paradigm used in Experiment 4 of Ciranka et al.'s study[5], and was programmed in PsychoPy 2022.2.2[26]. Seven images of everyday objects and animals drawn from the BOSS database[27] were randomly assigned a ground truth rank from 1-7 at the beginning of the experiment for each participant (Fig. 1A). Participants were told that their task was to learn about how the items related to one another with respect to how 'cnarcy' they are. They were informed that whether or not an item was more or less cnarcy than another was unrelated to any characteristics that these items have in real life. Since this nonsense word avoided any direct semantic associations that could influence participants' judgements, participants could only learn about cnarciness through the feedback provided on each trial of the experiment.

On each trial, following a 0.5 s fixation cross, two items were simultaneously presented on the left and right side of the screen on a white background for up to 2.5 s. Participants were instructed to select whichever item they thought was more cnarcy than the other as accurately and as quickly as possible using the left or right arrow key. They were informed that, on some trials, they would receive on-screen feedback about whether or not they had correctly chosen the cnarcier item (Fig. 1B). Unbeknownst to participants, the delivery of feedback was determined by the relative positions of the two items in the underlying cnarciness hierarchy that was established at the start of the experiment: if the two items were neighbouring in their rank ('adjacent trials', e.g. $i_3$ vs. $i_4$), then participants received feedback ("correct" or "incorrect") about their choice for 0.5 s, whereas if the items were non-neighbours ('TI trials', e.g. $i_3$ vs. $i_5$), then no feedback was provided. If no selection was made within 2.5 s, a 'missed response' was

recorded. Trials were separated by an inter-trial interval of 0.6 s. Each possible stimulus pairing ($N = 21$) was repeated with the left and right positions of items reversed. Each adjacent trial pairing ($N = 6$) was additionally repeated twice more (i.e. once with each L/R configuration), giving a total of 54 trials per block, of which 24 provided feedback and 30 provided no feedback. Trials with and without feedback were pseudo-randomly interleaved, allowing us to examine the evolution of TI over time. Concretely, we split each block into two shuffled sub-blocks within which each of the 21 possible item pairings (along with each of the six repeated adjacent pairs) was observed before being flipped and presented again in the second sub-block.

The entire experiment consisted of six blocks, each followed by a short attention check. At the start of the experiment, participants were informed that "*Not all items will stay as cnarcy for the entire experiment. Rather, on some blocks, certain items may (or may not) change in terms of how cnarcy they are. In other words, they may become more or less cnarcy, meaning their relations to the other items may change. So, keep aware of any changes in the feedback you receive!*". In addition, before each block, participants were reminded of the possibility that "*certain items may (or may not) change how cnarcy they are*". In reality, such a change was only introduced in the fourth block, such that from this block onwards, the ground truth item hierarchy changed in a manner determined by the group to which participants had been assigned. In the 'up' group, the hitherto lowest-ranking item $i_1$ moved 'up' the hierarchy to become the highest-ranking item, whereas in the 'down' group, the hitherto highest-ranking item $i_7$ moved 'down' the hierarchy to become the lowest-ranking item (Fig. 1A). In both groups, the relations between all other items remained exactly as they were before, such that the new ranking of item-IDs from lowest to highest could be represented as *7123456* in the 'down' group, and *2345671* in the 'up' group. Given that choice feedback continued to only be delivered on trials comparing items of neighbouring rank, such changes in ground truth structure resulted in the following minimal changes to the feedback received by each group: 1) participants in both groups now received feedback informing them that $i_7 < i_1$, 2) participants in the 'up' group now no longer received feedback that $i_1 < i_2$, and 3) participants in the 'down' group now no longer received feedback that $i_6 < i_7$. Thus, the objective changes in the underlying hierarchy could only possibly be inferred on the basis of the newly introduced $i_7 < i_1$ relation, and the persistence or omission of the $i_1 < i_2$ or $i_6 < i_7$ relation.

After the final block, participants performed three short tasks to test their explicit knowledge of the item hierarchy. First, using the mouse to drag and drop items, participants were asked to arrange the items according to how cnarcy they thought they were by the end of the experiment. Next, they were asked to click on whichever items (if any) they believed had changed how cnarcy they were at any point in the experiment. Finally, participants were given the opportunity to enter, using the keyboard, any comments they wanted to share on, for example, how they performed the task, the nature of the feedback received, or how difficult they thought the task was. While we do not formally analyse this post-task data here, we do note that most participants described the task in terms of general ease or difficulty, with some participants describing their strategy for solving the task (e.g., "creating a sort of scale in my head"). A few participants made more explicit references to the relational change in block 4, with one participant even correctly identifying that only one item had changed and noting the direction of change (e.g., "I noticed the fedora going from most cnarcy to least…"). These comments suggest that while much of the asymmetry bias may have been implicit, some participants were nonetheless consciously aware of the relational structure and its change. These post-task data are included in the open data repository (see Data Availability).

### Participants

Participants ($N = 150$) aged between 18–40 years were recruited online via Prolific Academic (74 female, 74 male, 2 non-binary; mean age 27 ± 5.27 years SD). After confirming their written informed consent, participants were randomly allocated to one of two groups: the 'up' group ($N = 76$; 37 female, 38 male, 1 non-binary; mean age = 27.14 ± 5.12 years SD), or the

'down' group ($N = 74$; 37 female, 36 male, 1 non-binary; mean age = $26.85 \pm 5.41$ years SD). Gender was self-reported by participants. We did not collect data on race/ethnicity. Participants received compensation of £6.00, plus a performance-dependent bonus of £2. The study was approved by the Ethics Committee of the Max Planck Institute for Human Development, and was not pre-registered.

Since our study focused on the impact of the changepoint manipulation on learned knowledge, we implemented a performance-related inclusion criterion. Participants in both groups experienced the same trial structure before the changepoint was reached (albeit with different randomised item allocations and trial sequences). We therefore used a binomial test to compute a performance threshold above which the likelihood that participants were performing at chance on *pre-changepoint trials* was 0.01 (i.e., following the criteria used by Ciranka et al.[5]), thus avoiding a confound by the experimental manipulation of interest. One additional participant was excluded for exhibiting a high proportion of missed responses (>60% of 322 trials). After the application of these criteria, $N = 83$ participants (36 female, 46 male, 1 non-binary; mean age = $26.90 \pm 5.34$ years SD) remained for analysis ('up': $N = 39$; 'down': $N = 44$).

## Behavioural models

Following Ciranka et al.[5], we modelled relational learning in our TI paradigm using simple RL models that updated the value (i.e., 'cnarciness') estimates $Q$ of winning and losing items $x$ and $y$, respectively, following choice feedback under a modified Rescorla-Wagner updating rule[28]:

$$Q_{t+1}(x) = Q_t(x) + \alpha^+[1 - d_t(x, y) - Q_t(x)] \tag{1}$$

$$Q_{t+1}(y) = Q_t(y) + \alpha^-[-1 + d_t(x, y) - Q_t(y)] \tag{2}$$

where $\alpha^+$ and $\alpha^-$ are the learning rates for winners and losers, respectively. Separating these learning rates allowed the model to implement varying degrees of symmetry/asymmetry in its learning updates. We defined the symmetric model *Q-symm* as an agent for whom $\alpha^+ = \alpha^-$, meaning that choice outcomes increased and decreased the agent's value estimates for winners and losers by equal amounts. In contrast, we defined the asymmetric model *Q-asymm* as an agent whose learning rates $\alpha^+$ and $\alpha^-$ could freely vary. In the case where $\alpha^+ > \alpha^-$, the agent was 'winner-biased', disproportionately increasing its value estimate for a comparison's winner relative to its loser, whereas the agent was 'loser-biased' if $\alpha^+ < \alpha^-$. This allowed us to obtain each participant's model-estimated asymmetry index $A$, calculated as the normalised difference between best-fitting learning rates under *Q-asymm*:

$$A = \frac{\alpha^+ - \alpha^-}{|\alpha^+ + \alpha^-|} \tag{3}$$

In the learning Eqs. 1,2, $d_t(x, y)$ represents the relative difference between $Q_t(x)$ and $Q_t(y)$, i.e.:

$$d_t(x, y) = \eta[Q_t(x) - Q_t(y)] \tag{4}$$

where $\eta$ is a scaling factor. This formalises the assumption that value updates scale with the difference between estimated item values. For instance, if an agent observes that $i_x > i_y$, this outcome should only induce a small change in value estimates for these items if the agent had already learned to expect this outcome (i.e., if $Q(x) >> Q(y)$). In contrast, observing that $i_x < i_y$ would be highly surprising, given the agent's current estimates of the relative values of these items, thus demanding a stronger update in the relevant value estimates. Incorporating such relational difference-weighting of value updates is necessary for *Q-symm* and *Q-asymm* to accomplish TI for non-anchor items (i.e., those of intermediate rank)[5]. We note that, depending on the value of the scaling factor $\eta$, the inclusion of the relative difference term $d_t$ can 'overflow' the bounds (i.e. 1 and -1) of the Rescorla-Wagner rule in Eqs. 1,2 – that is, the term that is added to $Q(x)$ in Eq. 1 or

subtracted from $Q(y)$ in Eq. 2. may end up being negative or positive, respectively. In such cases, the estimate of the winner would therefore *decrease*, and/or the estimate of the loser would *increase*. In order to prevent such edge cases, we therefore apply a positive rectifier function to the winner update and a negative rectifier function to the loser update, such that any negative winner updates or positive loser updates are clipped at 0.

Finally, we used a logistic choice function to define the probability of choosing $i_x > i_y$ based on the difference between estimated item values:

$$p_t(x > y) = \frac{1}{1 + e^{-(Q_t(x) - Q_t(y))/\tau}} \tag{5}$$

where $\tau$ is the temperature parameter determining the shape of the sigmoid function, and thus the degree of noise in choices based on item value differences. We note that Ciranka et al.[5] found that incorporating an additional 'pair-level' learning component, which allowed for episodic learning of pairs for which feedback was delivered, improved model fit by capturing enhanced learning for direct comparisons between neighbouring items. In the present study, we chose to omit this component to simplify the model space and focus specifically on how asymmetric learning, and the adaptability it confers, shapes inferential performance across transitive hierarchies.

The learning rates $\alpha^+$ and $\alpha^-$ remain static for the entire experiment for *Q-symm* and *Q-asymm*. To test whether learning asymmetries may differ as a function of the changepoint, we introduced *Q-asymm²*, which additionally possesses a separate pair of winner and loser learning rates for the pre- and post-changepoint phases respectively. In other words, the model initially learns using $\alpha^+_{pre}, \alpha^-_{pre}$, before using $\alpha^+_{post}, \alpha^-_{post}$ once the first $i_7 < i_1$ trial feedback is received.

We also tested another adaptive model, *Q-adapt*, capable of modulating the degree to which learning updates are shared between a given comparison's winner and loser on a trial-by-trial basis. On adjacent trials, we calculate an asymmetry modulator $\lambda$, bound between 0 and 1, as a quadratic function of the strength of the agent's prior belief about how items $x$ and $y$ are related upon receipt of choice feedback:

$$\lambda_t = -4\omega p_t(x > y)^2 + 4\omega p_t(x > y) + 1 - \omega \tag{6}$$

The value of $\lambda$ is minimal, causing more symmetric updating, when an agent's choice preference is strong and thus clearly supported or contradicted by the receipt of binary feedback (i.e. when $p(x > y)$ approaches 1 or 0), whereas it is maximal, causing more asymmetric updating, when the agent has no preference (i.e. when $p(x > y) = 0.5$). $\omega$ is an additional sensitivity parameter bound between 0 and 1 controlling the shape of the quadratic asymmetry modulator function (Fig. S5A). This determines how readily an agent adapts their degree of learning rate asymmetry as a function of the strength of their choice preference, effectively implementing a quadratic function that can be shallower or steeper depending on the value of $\omega$. When $\omega$ is 0, the agent's asymmetry is insensitive to changes in preference strength, such that $\lambda = 1$ (i.e., full asymmetric updating) for all choice probabilities. When $\omega$ is 1, Eq. 6 becomes roughly equivalent to a choice entropy function (cf. Equation 9 in Supplementary Note 1).

The $\lambda$ term can then be used to distribute the agent's base learning rate $\alpha^0$ across $\alpha_t^+$ and $\alpha_t^-$:

$$\alpha_t^+ = a^0 \frac{1 + \lambda_t}{2} - \begin{cases} a^0, & \text{if } a^0 < 0 \\ 0, & \text{otherwise} \end{cases} \tag{7}$$

$$\alpha_t^- = a^0 \frac{1 - \lambda_t}{2} - \begin{cases} a^0, & \text{if } a^0 < 0 \\ 0, & \text{otherwise} \end{cases} \tag{8}$$

Since $\alpha^0$ can take on negative values, the inclusion of the rightmost term in Eqs. 7 and 8 enables agents to vary in terms of whether their distribution of $\alpha^0$ across $\alpha^+$ and $\alpha^-$ is winner-biased (i.e. $\alpha^0 > 0$, and hence $\alpha_t^+ > \alpha_t^-$) or loser-biased (i.e. $\alpha^0 < 0$, and hence $\alpha^+ < \alpha^-$). Note that we assume that

*Q-adapt* cannot *reverse* its bias for the winners or losers of comparisons – for instance, for a winner-biased participant fit with $\alpha^0 > 0$, $\alpha_t^+$ can only be greater then or equal to $\alpha_t^-$. This is consistent with recent empirical work finding no evidence for an adaptive reversal of the sign of humans' learning asymmetries[23,24] (but see ref. 25).

## Model fitting and comparison

We estimated model parameters by minimising the negative log-likelihood of each participant's single-trial responses, given each model. Note that in all model fits and simulations, we directly evaluated the model-predicted choice probabilities (Eq. 5) to avoid binomial sampling error associated with drawing concrete (binary) model choices. We used Scipy's differential evolution method[29,30] over 500 iterations with the following lower and upper parameter bounds: $\alpha^+/\alpha^-$: (0;0.5); $\alpha^0$: (-0.5;0.5); $\eta$: (0;10); $\tau$: (0;1). From the resulting log-likelihood values under these best-fitting parameter estimates, we computed Akaike information criterion (AIC) values as an approximation of model evidence, where lower AIC indicates better goodness of fit, while penalising for model complexity[31]:

$$AIC = -2\log\left(p\left(D|M, \hat{\theta}\right)\right) + 2k \tag{9}$$

This amounts to the likelihood of a participant's choice data $D$, given a particular model $M$ and its best-fitting parameters $\hat{\theta}$, plus a penalty term $k$ corresponding to the number of free parameters. AIC values were, in turn, used to quantify the protected exceedance probability (pxp) associated with each competing model using the Variational Bayesian Analysis toolbox in MATLAB[32]. pxp values reflect the probability that a given model fits participants' data better than all other competing models, and hence is the most frequent data-generating model in the studied population. In contrast to the exceedance probability metric, the pxp additionally accounts for the null hypothesis that there is no difference in the frequencies of each model type[33].

To validate our model comparison approach and parameter inference, we performed model and parameter recovery analyses, revealing reliable identifiability of our candidate models and their fitted parameters, the details of which are outlined in 'Model and Parameter Recovery' in *Supplementary Methods* (Figs. S4A-D and S10A-B).

## Statistical tests

All statistical tests were performed as two-tailed tests. Behavioural effects, such as differences in accuracy or choice preference above chance, between groups, or across changepoints, were assessed using *t*-tests and ANOVAs, for which we used Cohen's *d* and $\eta_p^2$ as measures of effect sizes, respectively. 95% confidence intervals for values of $\eta_p^2$ were computed using an online tool: https://effect-size-calculator.herokuapp.com/. Data distribution was assumed to be normal, although this was not formally tested. In contrast, differences in model fit were tested using non-parametric Wilcoxon signed-rank tests, while differences in model parameters were tested using Mann-Whitney U-tests, owing to the non-normal distribution of AIC values and recovered learning rates. Non-parametric effect sizes were calculated as:

$$r = \frac{z}{\sqrt{N}} \tag{10}$$

where $z$ is the relevant non-parametric test's *z*-statistic, and $N$ is the sample size. To approximate the confidence interval for each non-parametric test's effect size, we used a bootstrapping approach with 10,000 samples. The threshold for significance in each case was 0.05, and a Bonferroni correction was applied in cases where post-hoc comparisons were deployed (e.g., for pairwise *t*-tests employed following significant ANOVA interaction effects).

## Reporting summary

Further information on research design is available in the Nature Portfolio Reporting Summary linked to this article.

## Results

### Simulations

We first present simulations of the symmetric and asymmetric RL agents *Q-symm* and *Q-asymm*, respectively, in order to derive model-based predictions for how humans should behave in our changepoint TI paradigm (Fig. 1C and S1). We simulated model performance over a range of parameter values matching those previously estimated to fit human TI behaviour by Ciranka et al.[5], where participants tended to exhibit a winner-biased learning rule (i.e., $\alpha^+ > \alpha^-$) when fitted with *Q-asymm*. Preferentially updating winners in this way leads to compression of *Q-asymm's* latent value structure before the changepoint, such that pairs of higher valued items are less discriminable than lower valued items. This reduced sensitivity towards larger values is a signature of asymmetry in relational learning. In contrast, the symmetric agent *Q-symm* exhibits no such compression (for details, see ref. 5).

Interestingly, *Q-asymm's* asymmetric learning predicts a difference in how efficiently it should adapt to our changepoint manipulation in the 'up' condition relative to the 'down' condition (Figs. 1C and S1). If learning is biased towards winners, the changepoint in the 'up' condition should be easily accommodated, since *Q-asymm* selectively and appropriately increases its value estimate for $i_1$ without needing to update any other items. On the other hand, in the 'down' condition, *Q-asymm's* initial tendency to increase its estimate for $i_1$ over-inflates this item's value, and underestimates $i_7$'s decline in value. In contrast, *Q-symm's* proportionate updating of winners and losers means that it will adapt to these two objective changes in the underlying ground truth with equal efficiency. Thus, if inferential learning is characterised by an asymmetric, winner-biased learning rule, then this yields the empirical prediction that humans should more efficiently adapt to the change in relational structure in the 'up' condition than in the 'down' condition.

### Value compression

To test these model predictions, we turned to the empirical data from our behavioural experiment. Focusing first on participants' pre-changepoint behaviour (that is, all trials preceding the first $i_7 < i_1$ trial in the fourth block), we confirmed that participants not only learned the cnarciness relations between items of neighbouring rank, but also used the feedback from these trials to accomplish TI (Fig. 2, leftmost column). Participants in both groups exhibited above-chance accuracy both on pre-changepoint trials involving adjacent items ('up': mean accuracy = $0.67 \pm 0.01$ SE, $t(38) = 12.94$, $p < 0.001$, $d = 2.07$, 95% CI = (0.15, 0.20); 'down': mean accuracy = $0.65 \pm 0.01$ SE, $t(43) = 10.35$, $p < 0.001$, $d = 1.56$, 95% CI = (0.12, 0.18)), and on pre-changepoint TI trials ('up': mean accuracy = $0.73 \pm 0.02$ SE, $t(38) = 13.22$, $p < 0.001$, $d = 2.12$, 95% CI = (0.20, 0.27); 'down': mean accuracy = $0.75 \pm 0.01$ SE, $t(43) = 16.67$, $p < 0.001$, $d = 2.51$, 95% CI = (0.22, 0.28)). In both groups, we also found evidence for the widely observed 'symbolic distance effect'[34,35] in both pre-changepoint accuracy and reaction time (RT) data, such that greater ordinal distance between comparanda on TI trials was associated with higher accuracy ('up': $\beta = 0.04$, $t(38) = 8.30$, $p < 0.001$, 95%, CI = (0.03, 0.05); 'down': $\beta = 0.05$, $t(43) = 11.07$, $p < 0.001$, 95% CI = (0.04, 0.06)) and faster responses ('up': $\beta = -0.03$, $t(38) = -4.11$, $p < 0.001$, 95% CI = $(-0.04, -0.02)$; 'down': $\beta = -0.03$, $t(43) = -5.24$, $p < 0.001$, 95% CI = $(-0.04, -0.02)$).

We next examined the extent to which participants' choice behaviour in the pre-changepoint period may have been reflective of a compressed latent value structure, a key signature of asymmetric learning. Inspecting participants' pairwise choice matrices (Fig. 3A, left panels) showed evidence of value compression, such that lower-valued TI pairs (that is, pairs of items closer towards the top-left corner of the choice matrix) tended to be judged more accurately than higher-valued TI pairs (that is, pairs of items closer towards the bottom-right corner of the choice matrix). We quantified the slope of this compression effect using linear regression (Fig. 4A and Fig. S2). Participants in both groups tended to exhibit asymmetry slopes significantly below 0, such that increases in combined pair value on TI trials were associated with a decline in accuracy ('up': mean $\beta = -0.02 \pm 0.01$ SE,

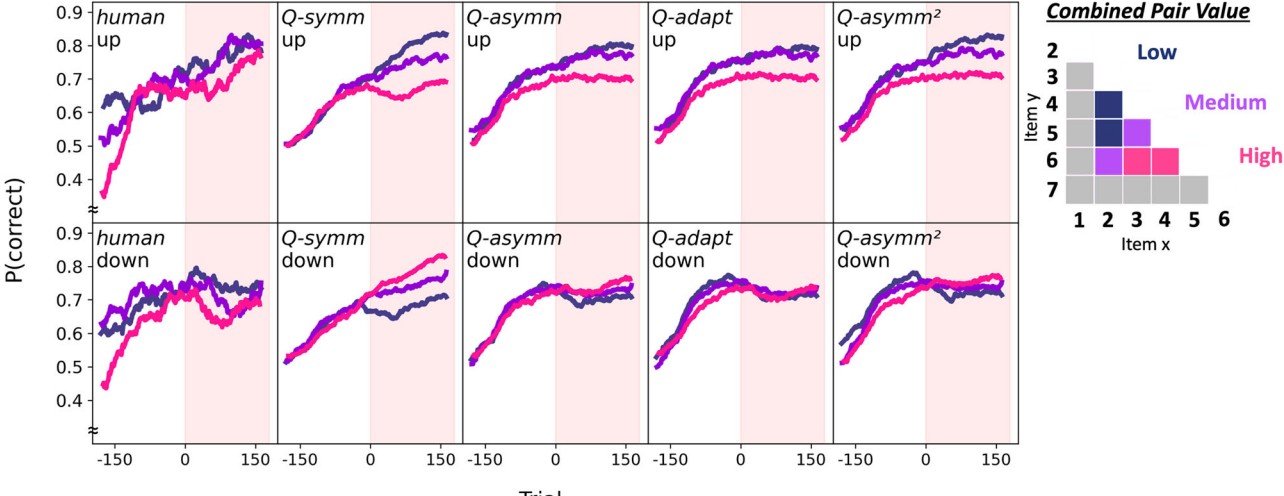

**Fig. 2 | TI accuracy over the course of the experiment in humans and fitted models.** Mean accuracy for TI pairs was calculated using a sliding window of 70 trials, where the x-axis refers to the trial relative to the first post-changepoint trial. Red shaded half of each panel represents the post-changepoint phase of the experiment. Dark blue, purple and pink colours respectively refer to low, medium, and high-valued TI comparisons, excluding anchors (see choice matrix in legend). Humans (leftmost column) exhibited a differential impact of the changepoint on TI

performance: whereas accuracy continued to improve in the 'up' group ($n = 39$; upper leftmost panel), post-changepoint accuracy was disrupted in the 'down' group ($n = 44$; lower leftmost panel). Simulating each candidate model using each participant's best-fitting parameters revealed that whereas the asymmetric model Q-asymm and the two adaptive models Q-adapt and Q-asymm² (third, fourth and fifth columns, respectively) qualitatively reproduced this interaction effect, the symmetric model Q-symm performed equally well in both conditions (first column).

$t(38) = -3.39$, $p = 0.002$, $d = -0.54$, 95% CI = $(-0.04, -0.01)$; 'down': mean $\beta = -0.02 \pm 0.01$ SE, $t(43) = -3.62$, $p < 0.001$, $d = -0.55$, 95% CI = $(-0.03, -0.01)$). This degree of asymmetry did not significantly differ between groups ($t(81) = 0.10$, $p = 0.918$, $d = 0.02$, 95% CI = $(-0.02, 0.02)$). In line with previous work, we therefore found evidence that during the initial pre-changepoint phase, participants acquired a compressed value structure, consistent with an asymmetric learning strategy.

**Differential impact of changepoint on TI performance**

Turning to post-changepoint behaviour, we examined how effectively participants accommodated the different shifts in the ranking of one of the anchor items (i.e., $i_1$ or $i_7$) while preserving their knowledge about the remaining items (Fig. 2, leftmost column). To isolate the impact of each changepoint on downstream inferential knowledge, and to avoid any skewing effect of pre-changepoint preferences for the moved anchor items, we focused on comparisons involving non-anchor items whose rank position had not changed in either group (i.e., from $i_2$ to $i_6$). Post-changepoint non-anchor accuracy was significantly above chance in both groups for adjacent pairs ('up': mean accuracy = $0.80 \pm 0.02$ SE, $t(38) = 12.10$, $p < 0.001$, $d = 1.94$, 95% CI = $(0.25, 0.35)$; 'down': mean accuracy = $0.73 \pm 0.03$ SE, $t(43) = 8.48$, $p < 0.001$, $d = 1.28$, 95% CI = $(0.17, 0.28)$), and for TI pairs ('up': mean accuracy = $0.74 \pm 0.03$ SE, $t(38) = 7.03$, $p < 0.001$, $d = 1.13$, 95% CI = $(0.17, 0.31)$; 'down': mean accuracy = $0.70 \pm 0.03$ SE, $t(43) = 6.37$, $p < 0.001$, $d = 0.96$, 95% CI = $(0.14, 0.27)$). To evaluate how accuracy developed from one phase of the experiment to the next, and whether these effects differed between groups, we conducted a series of $2 \times 2$ mixed ANOVAs with changepoint (pre vs. post) as a within-subjects factor, and direction ('up' vs. 'down') as a between-subjects factor. For adjacent pairs, we observed a significant main effect of changepoint, reflecting a significant increase in accuracy from the first to the second half of the experiment (pre-changepoint: mean accuracy = $0.62 \pm 0.01$ SE; post-changepoint: mean accuracy = $0.76 \pm 0.02$ SE; $F(1,81) = 83.13$, $p < 0.001$, $\eta_p^2 = 0.51$, 95% CI = $(0.35, 0.61)$). Adjacent trial accuracy did not significantly differ between direction groups across the whole experiment ('up': mean accuracy = $0.72 \pm 0.02$ SE; 'down': mean accuracy = $0.67 \pm 0.02$ SE; $F(1,81) = 3.87$, $p = 0.053$, $\eta_p^2 = 0.05$, 95% CI = $(0.00, 0.16)$), nor was there a significant changepoint x direction interaction effect ($F(1,81) = 1.25$, $p = 0.268$,

$\eta_p^2 = 0.02$, 95% CI = $(0.00, 0.10)$). Repeating this $2 \times 2$ ANOVA on TI accuracy, we likewise observed a main effect of changepoint, which was similarly driven by an improvement in TI accuracy from the pre- to the post-changepoint phase of the experiment (pre-changepoint: mean accuracy = $0.65 \pm 0.02$ SE; post-changepoint: mean accuracy = $0.72 \pm 0.02$ SE; $F(1,81) = 20.00$, $p < 0.001$, $\eta_p^2 = 0.20$, 95% CI = $(0.06, 0.34)$). While the main effect of direction on TI trial accuracy was non-significant ('up': mean accuracy = $0.68 \pm 0.03$ SE; 'down': mean accuracy = $0.69 \pm 0.03$ SE; $F(1,81) < 0.01$, $p = 0.950$, $\eta_p^2 < 0.01$, 95% CI = $(0.00, 0.02)$), we observed a significant changepoint x direction interaction ($F(1,81) = 5.87$, $p = 0.018$, $\eta_p^2 = 0.07$, 95% CI = $(0.00, 0.19)$). Bonferroni-corrected post-hoc comparisons revealed that while participants in the 'up' group exhibited a significant improvement in TI accuracy from the pre- to the post-changepoint phases (pre-changepoint: mean accuracy = $0.63 \pm 0.03$ SE; post-changepoint: mean accuracy = $0.74 \pm 0.03$ SE; $t(38) = 5.19$, $p < 0.001$, $d = 0.83$, 95% CI = $(0.07, 0.15)$), participants in the 'down' group showed no such effect (pre-changepoint: mean accuracy = $0.67 \pm 0.02$ SE; post-changepoint: mean accuracy = $0.70 \pm 0.03$ SE; $t(43) = 1.51$, $p = 0.277$, $d = 0.23$, 95% CI = $(-0.01, 0.08)$).

To further probe how the changepoint may have differentially impacted inferential learning in the two groups, we fitted participants' post-changepoint non-anchor TI trials using a logistic mixed-effects model, with trial number and changepoint direction as metric and categorical predictors, respectively. Participant-specific random intercepts and random slopes for trial number were included to account for individual differences in overall accuracy and learning trajectories. This revealed a significant main effect of trial number, indicating that participants' accuracy improved over time after the changepoint ($\beta = 0.32$, 95% CI = $(0.12, 0.51)$, SE = $0.10$, $z = 3.21$, $p = 0.001$). The main effect of direction was not significant ($\beta = -0.32$, 95% CI = $(-0.97, 0.34)$, SE = $0.34$, $z = -0.95$, $p = 0.340$), suggesting no overall difference in accuracy between the two direction conditions. Importantly, we observed a significant interaction between trial and changepoint direction ($\beta = -0.28$, 95% CI = $(-0.52, -0.04)$, SE = $0.12$, $z = -2.25$, $p = 0.024$), consistent with the idea that participants in the 'down' group exhibited a shallower post-changepoint learning trajectory compared to those in the 'up' group. Together, this indicates that the changepoint manipulation differentially impacted participants' ability to infer transitive relations among unchanged items: while participants continued to improve non-

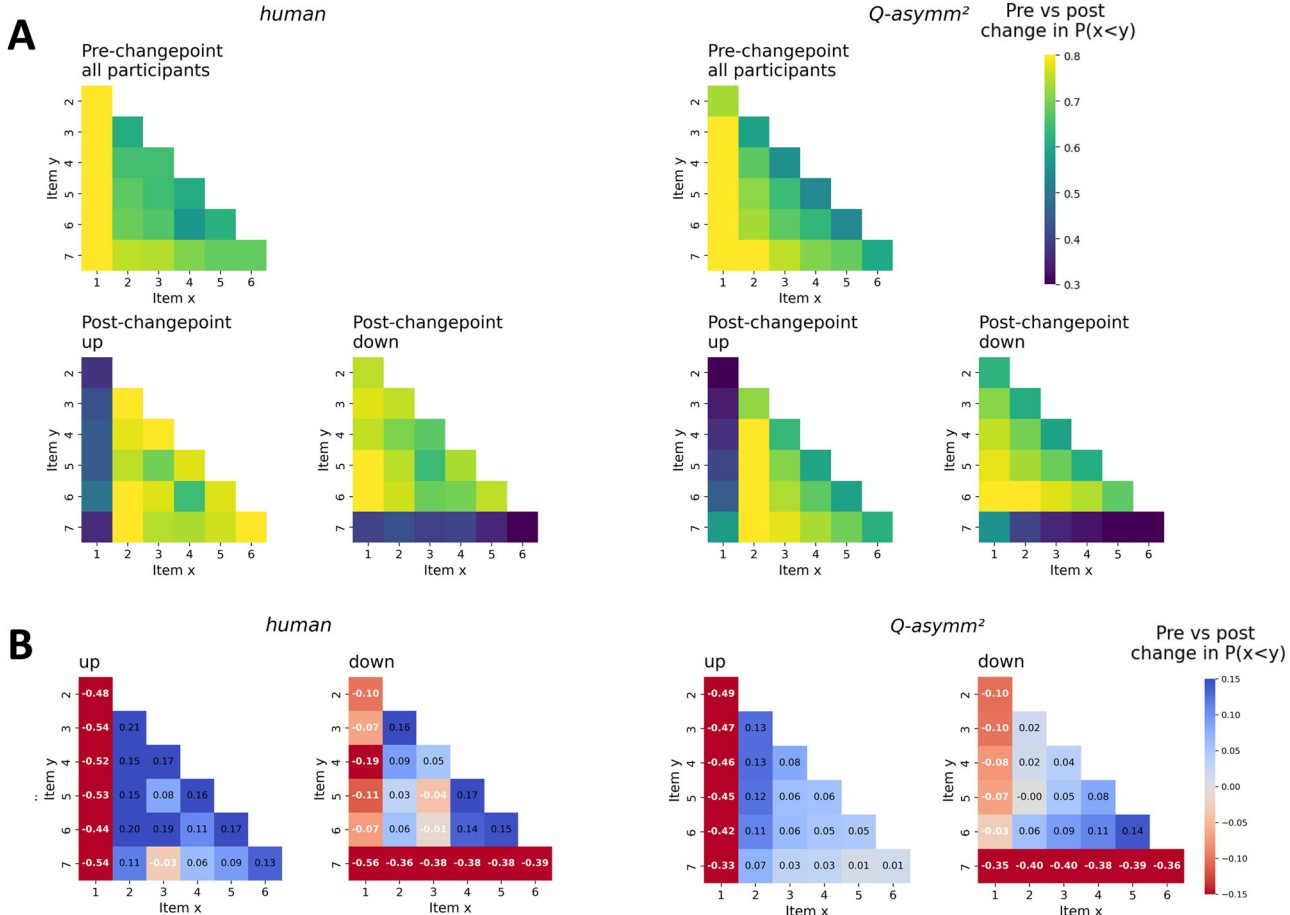

**Fig. 3 | Choice matrices for humans (left panels) and the best-fitting model *Q-asymm²* (right panels). A** P(x < y), i.e. the mean probability of choosing item y (matrix rows) over item x (matrix columns), as represented by the colour-bar. Top row of panels displays pre-changepoint data collapsed across 'up' (*n* = 39) and 'down' (*n* = 44) participants, while the bottom row of panels splits post-changepoint data by group. **B** Pre vs. post-changepoint change in P(x < y). Blue colours indicate that the agent's preference for item y over x has increased, while red colours indicate that it has decreased. Colour-bar value range was narrowed between 0.3 and 0.8 in Fig A and −0.15 and 0.15 in Fig B to improve legibility of differences among non-anchor pairs, while in-plot annotations in Fig B also indicate the values for these differences.

anchor TI accuracy when $i_1$ moved to the top of the hierarchy, non-anchor TI learning was relatively stunted in participants for whom $i_7$ moved to the bottom of the hierarchy.

We next investigated the extent to which participants appropriately switched their choice preferences for whichever anchor item had moved to the other end of the hierarchy after the changepoint – i.e. P(choose $i_1$) for 'up' participants, and P(choose $i_7$) for 'down' participants (Figure. S3; note: we excluded all feedback trials from this analysis, as well as all $i_1$ vs. $i_7$ trials, in order to isolate any changes in preference for these moved anchors with respect to non-neighbouring, non-anchor items). In 'up' participants, we observed a significant increase in participants' preference for the moved anchor $i_1$ after the changepoint (pre-changepoint: mean = 0.13 ± 0.02 SE; post-changepoint: mean = 0.57 ± 0.06 SE; *t*(38) = 6.74, *p* < 0.001, *d* = 1.08, 95% CI = (0.30, 0.56)), and likewise a significant decrease in 'down' participants' tendency to choose $i_7$ after the changepoint (pre-changepoint: mean = 0.72 ± 0.03 SE; post-changepoint: mean = 0.38 ± 0.05 SE; *t*(43) = −7.70, *p* < 0.001, *d* = −1.16, 95% CI = (−0.43, −0.25). The *absolute difference* in choice preferences for the moved anchor before and after the changepoint did not significantly differ between the two groups ('up': mean difference 0.43 ± 0.06 SE; 'down': mean difference = 0.34 ± 0.04 SE; *t*(81) = 1.16, *p* = 0.249, *d* = 0.26, 95% CI = (−0.06, 0.25)). Thus, both groups of participants appeared equally capable of correctly re-positioning whichever anchor item had moved to the other end of the hierarchy. Interestingly, however, we also observed differences in the changepoint's impact on participants' tendency to select the *unmoved* anchor item. Specifically,

whereas participants in the 'up' group did not significantly differ in their preference for the unmoved anchor $i_7$ before vs after the changepoint (pre-changepoint: mean = 0.72 ± 0.03 SE; post-changepoint: mean = 0.76 ± 0.04 SE; *t*(38) = 1.68, *p* = 0.102, *d* = 0.27, 95% CI = (−0.01, 0.10)), participants in the 'down' group exhibited a significant increase in their preference for the unmoved anchor $i_1$ (pre-changepoint: mean = 0.14 ± 0.02 SE; post-changepoint: mean = 0.21 ± 0.03 SE; *t*(43) = 2.22, *p* = 0.031, *d* = 0.34, 95% CI = (0.006, 0.13)). Together, these findings suggest that while both sets of participants appropriately adjusted their preferences for the moved anchor item, 'down' participants were additionally biased towards increasing their tendency to choose the bottom anchor $i_1$, even despite the fact that it had not changed its rank. Participants may therefore have displayed a certain degree of symmetric (or even reversed asymmetric) updating of the 'moved' anchor item itself after the changepoint, alongside the more general winner-biased asymmetric updating of all other items (including the unmoved anchor item), a possibility that we return to later in the *Results* section.

## Model asymmetry

The foregoing behavioural analyses suggest that participants exhibited value compression effects and differential sensitivity to changes in relational structure consistent with winner-biased belief updating. Next, we compared the relative fits of our symmetric and asymmetric RL models (*Q-symm* and *Q-asymm*) to the human experiment data. In both groups of participants, *Q-asymm* provided a better fit to participants' behaviour than *Q-symm*, as confirmed using Wilcoxon signed-rank tests of AICs (Fig. 5A; 'down': mean

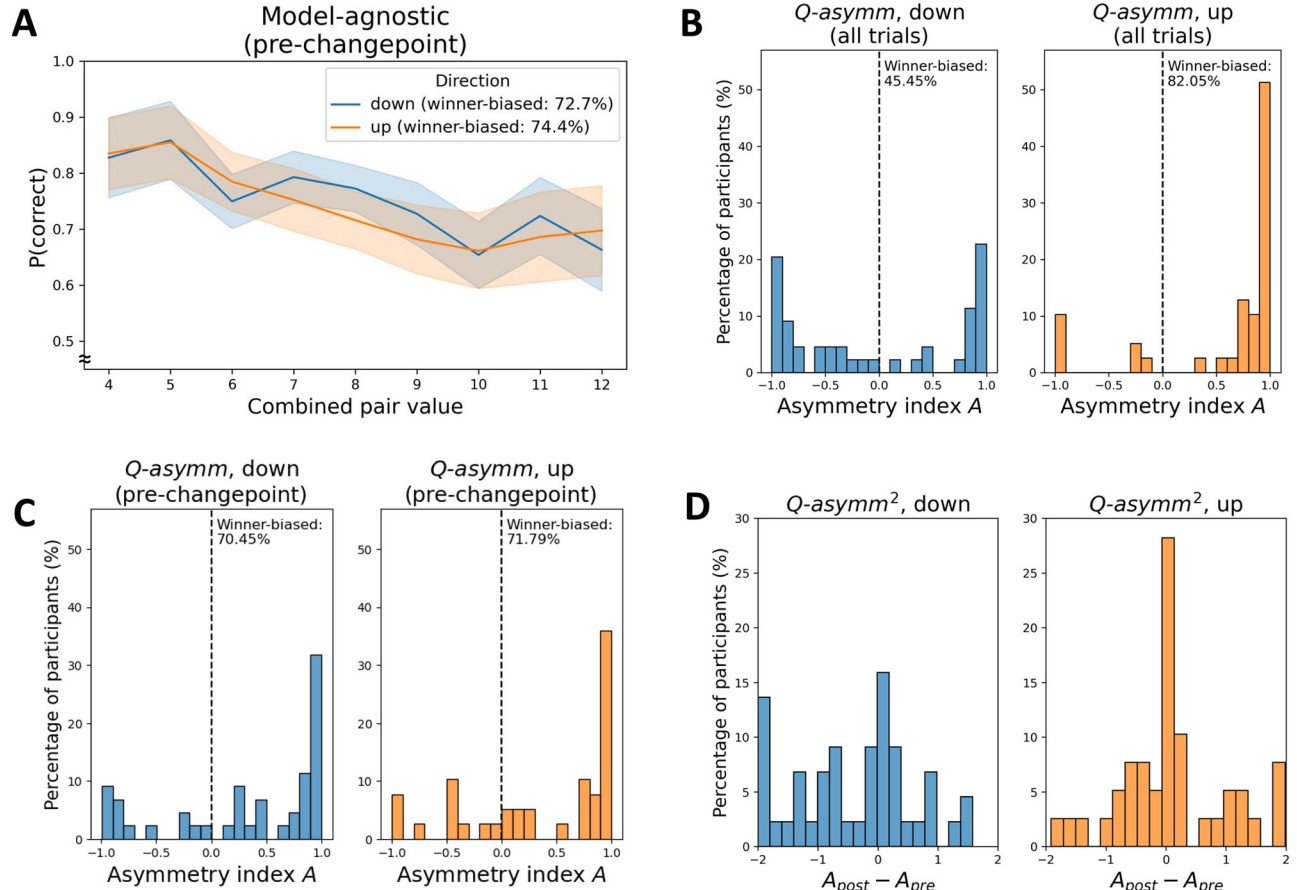

**Fig. 4 | Model-agnostic and model-estimated learning asymmetry. A** The model-agnostic measure of participants' learning asymmetry is the slope of the relationship between TI accuracy and combined pair value on all pre-changepoint trials (see also Fig. S2). Participants with negative slopes are designated as winner-biased (see in-text legend for percentages). Error bars reflect 95% confidence intervals for mean TI accuracy per combined pair value. **B** In contrast, the *Q-asymm*-based asymmetry measure refers to the normalised difference in best-fitting learning rates, where −1, 0 and +1 values for *A* indicate full loser bias, symmetry, and full winner bias, respectively. Whereas 'up' participants ($n = 39$) tended to be strongly winner-biased when *Q-asymm* was fit to trials from the whole experiment (i.e., strong left-skew in right panel), 'down' participants ($n = 44$) were estimated to be more evenly split between winner- and loser-biased (i.e., bimodal distribution in

left panel). The proportion of participants designated as winner- or loser-biased in the 'down' group according to this model-based metric therefore substantially deviated from that according to the model-agnostic metric in Fig. A (see in-plot percentages). **C** In contrast, *Q-asymm* models fit to pre-changepoint trials were predominantly winner-biased in both groups. **D** We fit *Q-asymm²* to participant data, which was equivalent to *Q-asymm*, except that its two learning rates reset after the changepoint. Calculating the difference between the model's pre- and post-changepoint asymmetry index *A* revealed a tendency to become less winner-biased in the 'down' group (left panel). This indicates that while participants initially exhibited a predominantly winner-biased learning asymmetry, the degree and sign of this asymmetry did not remain static over the course of the entire experiment.

*Q-asymm* AIC = 349.81 ± 10.46 SE; mean *Q-symm* AIC = 366.12 ± 9.71 SE; $z = 5.22$, $p < 0.001$, $r = 0.79$, 95% CI = (0.67, 0.85); 'up': mean *Q-asymm* AIC = 339.26 ± 12.48 SE; mean *Q-symm* AIC = 363.53 ± 10.99 SE; $z = 5.25$, $p < 0.001$, $r = 0.84$, 95% CI = (0.77, 0.87)). Comparison of pxps likewise revealed, in both groups of participants, a clear advantage for *Q-asymm* over *Q-symm* ('up': *Q-asymm* pxp > 0.99, *Q-symm* pxp < 0.01; 'down': *Q-asymm* pxp > 0.99, *Q-symm* pxp < 0.01). These initial model comparison analyses therefore not only replicate previously observed learning asymmetries, but also suggest that the differential impact of the changepoint in our modified TI setting is likewise better captured by a model furnished with asymmetric, rather than symmetric, learning rates.

We next examined the model-estimated asymmetry index *A* of each participant under the *Q-asymm* model, where values of *A* closer to 1 or −1 indicate greater winner or loser biases respectively, and $A = 0$ indicates perfect symmetry between learning rates (see Eq. 3. in *Methods*, 'Behavioural Models'). As in previous work[5], values of *A* tended to be left-skewed in the 'up' group, indicating a strongly winner-biased learning asymmetry (Fig. 4B, right panel). In addition to this majority of 'up' participants who were estimated to be winner-biased ($N = 32/39$), there was also a small sub-group of participants for whom *A* was lower than 0, and hence who were

estimated to be loser-biased under the best-fitting *Q-asymm* model ($N = 7/39$). In contrast, *A* values for 'down' participants exhibited a more starkly bimodal distribution, such that participants were modelled as more evenly split between being either strongly winner-biased ($N = 20/44$) or loser-biased ($N = 24/44$) (Fig. 4B, left panel). Indeed, non-parametric statistical comparisons revealed significantly lower values of *A* in 'down' participants compared to 'up' participants ('down': mean $A = -0.02 ± 0.12$ SE; 'up': mean $A = 0.62 ± 0.10$ SE; Mann-Whitney-U-test: $U = 1269$, $z = 3.75$, $p < 0.001$, $r = 0.41$, 95% CI = (0.21, 0.59)). In contrast, when fitting *Q-asymm* to participants' *pre-changepoint* choices only, we observed no significant difference in model-estimated asymmetry ('down': mean $A = 0.34 ± 0.11$ SE; 'up': mean $A = 0.38 ± 0.11$ SE; Mann-Whitney-U-test: $U = 876$, $z = 0.16$, $p = 0.873$, $r = 0.02$, 95% CI = (−0.19, 0.23)). The *A* values obtained from these pre-changepoint fits instead tended to be similarly left-skewed in both groups, providing estimates for the number of winner and loser-biased participants ('up': winner-biased $N = 28/39$, loser-biased $N = 11/39$; 'down': winner-biased $N = 31/43$, loser-biased $N = 13/43$) that more closely matched those obtained under our model-agnostic asymmetry slope metric reported earlier (Figs. 4C; cf. Fig. 4A and S2). This suggests that while participants' pre-changepoint behaviour may be best explained by a winner-

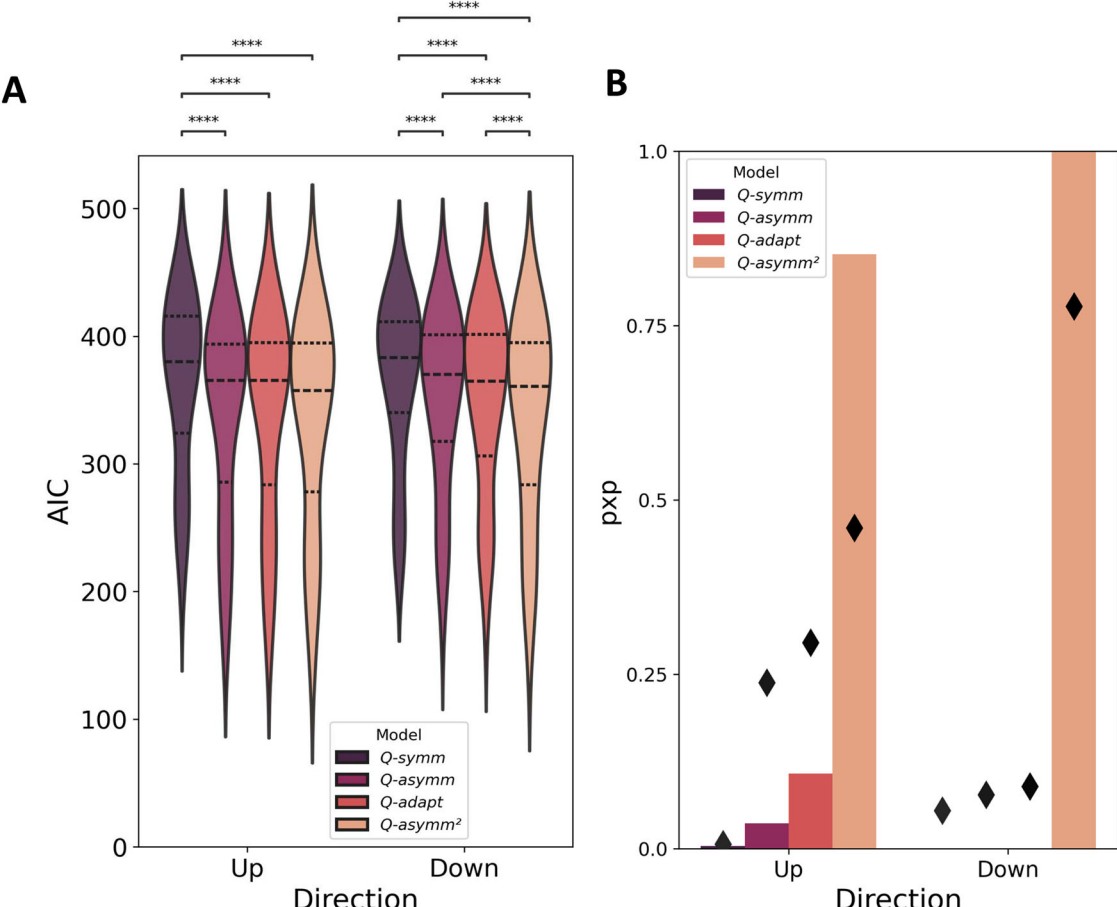

**Fig. 5 | Model comparison for each candidate model, within each task condition.**
**A** Lower AIC values indicate better fit of the model to the behavioural data. Dashed lines indicate quartiles of the data, while asterisks indicate a significant difference between AIC values for a given pair of models (i.e., $p < 0.05$; Wilcoxon signed-rank tests). **B** Higher pxp values (bars) indicate greater probability that a given model is the most frequent data-generating model in the studied population, while diamonds indicate the estimated frequency of each model – that is, the expected proportion of participants best described by each model.

biased learning rule, our model fitting procedure may have biased *Q-asymm*-derived learning rates towards capturing post-changepoint behaviour, when in fact this degree of bias may have varied across task phases. We address this possibility in the following section.

**Adaptive asymmetry**

Our original hypothesis was that a differential impact of the changepoint on TI performance would arise as a direct consequence of the agent's asymmetric learning rule – that is, the relative ease (or difficulty) in accommodating the 'up' (or 'down') relational change should be a function of each agent's tendency to preferentially update winners or losers. Such hypotheses were therefore derived under the assumption of a static degree of asymmetry, whereby each agent's preferential updating of winners (or losers) remained constant over the course of the task, even in the face of the changepoint. However, it is also important to consider the possibility that such asymmetries may have varied over time as learning progressed, or as a function of objective changes in the task – namely, the changepoint.

To evaluate the possibility that participants' degree of learning asymmetry may have differed before and after the changepoint, we fitted a variant of *Q-asymm* – i.e., *Q-asymm²* – equipped with two separate pairs of learning rates for winners and losers for each experimental phase, i.e., $\alpha^+_{pre}$, $\alpha^-_{pre}$ and $\alpha^+_{post}$, $\alpha^-_{post}$. We also considered an adaptive agent *Q-adapt*, whose degree of asymmetry varied on a trial-by-trial basis as a function of the strength of the agent's preference regarding the cnarciness relation between the two compared items – i.e., the probability of choosing one item over the other (see Eqs. 6–8, Fig. S5A–C and Supplementary Note 1). We formalised this more dynamic model in light of previous evidence that the sparsity of feedback

modulates learning asymmetry[5]. In the present case, the rationale was that trials for which the agent's preference for one of the two items is weaker may induce them to (asymmetrically) allocate a larger proportion of the overall update to one of the items. In contrast, on trials where the agent has a stronger preference, the receipt of feedback should provide a clear indication that this prior belief needs to be further reinforced or reversed via a more symmetrically distributed updating of both items. Thus, we considered two alternative candidate models whose adaptability manifested itself differently in each case: whereas *Q-asymm²* exhibited a single reset in learning rates upon receipt of changepoint-indicative feedback, *Q-adapt* dynamically modulated its learning asymmetry as a function of changes in preference strength.

After fitting *Q-asymm²* and *Q-adapt* to participants' choices over the whole experiment, we repeated the Bayesian model selection steps to calculate model pxps, given the inclusion of these additional models (Fig. 5A, B). Among both groups of participants, the dynamic model *Q-adapt* yielded a slight but non-significant improvement in terms of AIC over *Q-asymm* ('up': *Q-adapt*: mean AIC = 339.20 ± 12.65 SE; *Q-asymm*: mean AIC = 339.26 ± 12.48 SE; Wilcoxon signed-rank test of AICs: $z = 0.27$, $p = 0.791$, $r = 0.04$, 95% CI = (0.00, 0.37); 'down': *Q-adapt*: mean AIC = 348.37 ± 10.60 SE; *Q-asymm*: mean AIC = 349.81 ± 10.46 SE; Wilcoxon signed-rank test of AICs: $z = 0.81$, $p = 0.421$, $r = 0.12$, 95% CI = (0.01, 0.41)). *Q-asymm²* provided a more convincing improvement in model fit, exhibiting a small but non-significant improvement in AIC relative to the next-best model *Q-adapt* in the 'up' group (*Q-asymm²*: mean AIC = 334.15 ± 13.04 SE; Wilcoxon signed-rank test of AICs: $z = 1.59$, $p = 0.112$, $r = 0.25$, 95% CI = (0.02, 0.53)), and a significant improvement in the 'down' group

**A**

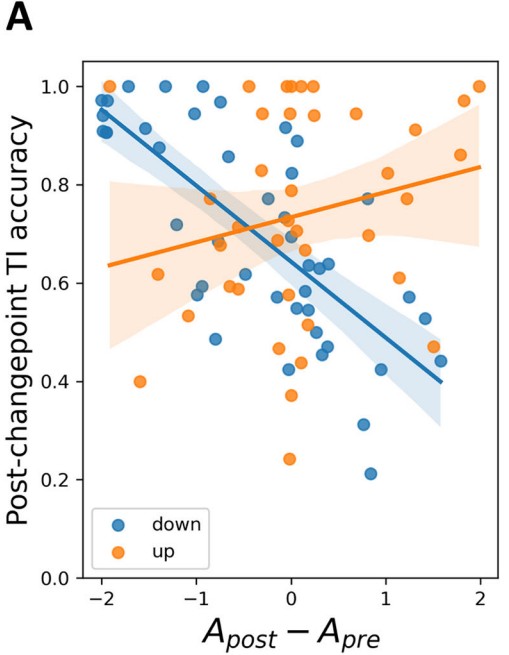

**B**

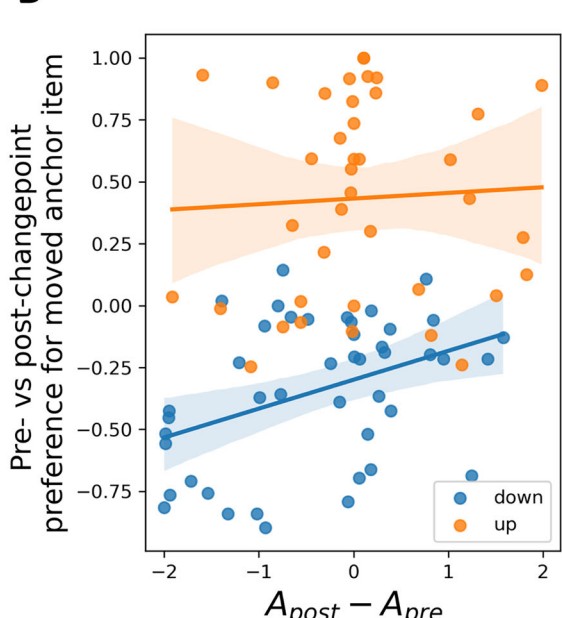

**Fig. 6 | Changepoint-induced changes in learning asymmetry are related to behavioural performance.** Among 'down' participants ($n = 44$), pre- vs. post-changepoint differences in learning asymmetry estimated with $Q\text{-}asymm^2$ were significantly negatively correlated with post-changepoint non-anchor TI accuracy (**A**), and positively correlated with their tendency to correctly change their preference for the moved anchor item (i.e., $i_7$; **B**). In other words, participants in the 'down' group who appropriately adapted to the change in relational structure exhibited a larger reduction in their winner-biased learning asymmetry. In contrast, these relationships were non-significant among 'up' participants ($n = 39$). Error bars reflect 95% confidence intervals for regression estimates.

($Q\text{-}asymm^2$: mean AIC = 337.20 ± 11.75 SE; Wilcoxon signed-rank test of AICs: $z = 4.60$, $p < 0.001$, $r = 0.69$, 95% CI = (0.52, 0.81)). Comparison of pxps revealed a more convincing advantage for $Q\text{-}asymm^2$ over other competitor models in both groups ('up': $Q\text{-}asymm^2$: pxp = 0.85; $Q\text{-}adapt$: pxp = 0.11; $Q\text{-}asymm$: pxp = 0.04; $Q\text{-}symm$: pxp < 0.01; 'down': $Q\text{-}asymm^2$: pxp > 0.99; $Q\text{-}adapt$: pxp < 0.01; $Q\text{-}asymm$: pxp < 0.01; $Q\text{-}symm$: pxp < 0.01). We note that under the Bayesian information criterion (BIC), a model evidence metric that more strongly penalises model complexity and thus favours more parsimonious models, $Q\text{-}adapt$ and $Q\text{-}asymm^2$ emerged as the winning models in the 'up' and 'down' groups respectively, although this metric led to worse model recovery than AIC (Fig. S4C, D and Fig. S6A, B). Together, this indicates that $Q\text{-}asymm^2$ outperformed both its static counterparts $Q\text{-}symm$ and $Q\text{-}asymm$ and the more dynamically adaptive model $Q\text{-}adapt$, particularly with respect to 'down' participants.

To further probe the nature of this shift in learning rates, we calculated the asymmetry indices $A_{pre}$ and $A_{post}$ of $Q\text{-}asymm^2$ using each of these pairs of fitted learning rates (Fig. 4D). Participants in the 'up' group showed a winner-biased learning asymmetry in the pre-changepoint phase that did not significantly differ between changepoints (mean $A_{pre} = 0.42 ± 0.11$ SE; mean $A_{post} = 0.51 ± 0.10$ SE; Wilcoxon signed-rank test: $z = 0.35$, $p = 0.727$, $r = 0.06$, CI = (0.00, 0.38)). However, participants in the 'down' group underwent a significant reduction in their winner-biased learning asymmetry after the changepoint (mean $A_{pre} = 0.37 ± 0.10$ SE; mean $A_{post} = -0.02 ± 0.12$ SE; Wilcoxon signed-rank test: $z = -2.04$, $p = 0.041$, $r = -0.31$, CI = (−0.56, −0.04)). Inspecting these differences in asymmetry across changepoints indicated a bimodal split of participants between those whose asymmetry indices remained stable across changepoints - i.e. their bias remained unchanged – and those who showed a pronounced reduction, or even a reversal in the sign, of their asymmetry index – i.e. a tendency towards becoming more symmetric, or even loser-biased (see Fig. 4D). Accordingly, we observed that best-fitting values for the $Q\text{-}adapt$'s adaptability parameter, $\omega$, were bimodally distributed around 0 and 1 (i.e. corresponding to no adaptability and maximal adaptability of learning asymmetry, respectively), and did not significantly differ between groups ('up': mean $\omega = 0.37 ± 0.06$ SE; 'down': mean $\omega = 0.47 ± 0.06$ SE; Mann-

Whitney U-test: $U = 664$, $z = 1.77$, $p = 0.077$, $r = 0.19$, 95% CI = (−0.03, 0.40)). This indicates that participants in both groups could be broadly divided into those whose learning asymmetry was or was not sensitive to changes in choice preference strength, in agreement with the distinction of adaptive and non-adaptive participants under $Q\text{-}asymm^2$.

Interestingly, the 'down' participants for whom this change in learning asymmetry was most pronounced tended to be those who exhibited relatively high post-changepoint performance. For instance, participants' difference between $A_{post}$ and $A_{pre}$ under $Q\text{-}asymm^2$ was significantly negatively correlated with their post-changepoint non-anchor TI accuracy, and hence with their capacity to respond to the changepoint while minimising disruption to the unchanged transitive hierarchy (Spearman's rank coefficient: $r = -0.73$, $p < 0.001$; Fig. 6A). Likewise, this reduction in learning asymmetry after the changepoint was positively correlated with participants' pre- versus post-changepoint change in preference for the moved anchor $i_7$, such that participants who correctly reduced their preference for $i_7$ tended to show a greater reduction in their winner-bias after the changepoint ($r = 0.37$, $p = 0.015$; Fig. 6B). In contrast, no such significant relationship held for 'up' participants, neither with respect to their post-changepoint non-anchor TI accuracy ($r = 0.21$, $p = 0.209$), nor their change in preference for the moved anchor $i_1$ ($r = 0.11$, $p = 0.487$). These findings lend further support to the idea that although the changepoint experienced by 'down' participants disrupted TI learning at the group level, well-performing participants were nonetheless capable of leveraging an adaptive reduction in winner-biased asymmetry to respond more appropriately to the change in ground truth.

**Model validation**

As a final model validation step, we simulated our candidate models using the best-fitting empirical parameters, and then analysed the resulting choice probabilities to verify which models were capable of qualitatively reproducing the key behavioural effects observed in our empirical dataset[36,37]. We first examined the consistency of the human and model-estimated value compression effects. In line with the descriptive results (cf. Fig. 3A, upper left panel), the pre-changepoint TI performance of both $Q\text{-}adapt$ and $Q\text{-}asymm^2$ was characterised by a compressed value structure, with

asymmetry slopes significantly below 0 ($Q\text{-}asymm^2$: 'up': mean $\beta = -0.02 \pm 0.005$ SE, $t(38) = -3.96$, $p < 0.001$, $d = -0.63$, 95% CI $= (-0.03, -0.01)$; 'down': mean $\beta = -0.02 \pm 0.004$ SE, $t(43) = -4.54$, $p < 0.001$, $d = -0.68$, 95% CI $= (-0.03, -0.01)$; $Q\text{-}adapt$: 'up': mean $\beta = -0.02 \pm 0.01$ SE, $t(38) = -3.89$, $p < 0.001$, $d = -0.62$, 95% CI $= (-0.03, -0.01)$; 'down': mean $\beta = -0.01 \pm 0.01$ SE, $t(43) = -2.44$, $p = 0.019$, $d = -0.37$, 95% CI $= (-0.03, 0.00)$). We next identified participants as winner- or loser-biased according to their pre-changepoint asymmetry index $A_{pre}$ ($Q\text{-}asymm^2$), or the sign of their best-fitting $\alpha^0$ value ($Q\text{-}adapt$). Both models yielded estimates for the proportion of participants falling into each category that were more closely in line with those gleaned from the model-agnostic measure, i.e., the sign of participants' pre-changepoint asymmetry slope (number of winner-biased participants under $Q\text{-}asymm^2$: 'down': 32/44 participants; 'up': 28/39 participants; number of winner-biased participants under $Q\text{-}adapt$: 'down': 25/44 participants; 'up': 33/39 participants; cf. Fig. 4B and Fig. S2). This stands in contrast to $Q\text{-}asymm$, which failed to reproduce a significantly negative asymmetry slope among 'down' participants ('up': mean $\beta = -0.02 \pm 0.01$ SE, $t(38) = -4.03$, $p < 0.001$, $d = -0.64$, 95% CI $= (-0.03, -0.01)$; 'down': mean $\beta = -0.01 \pm 0.01$ SE, $t(43) = -1.44$, $p = 0.157$, $d = -0.22$, 95% CI $= (-0.02, 0.00)$), while also underestimating the proportion of winner-biased participants according to its model-based asymmetry index $A$, as reported earlier.

Turning to model behaviour as a function of the changepoint, we repeated our $2 \times 2$ mixed ANOVA on our models' non-anchor TI choice accuracy, with changepoint (pre vs. post) as a within-subjects factor, and direction ('up' vs. 'down') as a between-subjects factor. Interestingly, despite both emerging as the best model in terms of its fit to behavioural data, as well as appearing to qualitatively reproduce the changepoint-related changes in TI accuracy (Fig. 2, right-most panels), $Q\text{-}asymm^2$ failed to reproduce the significant changepoint x direction interaction effect observed in humans ($F(1,81) = 2.56$, $p = 0.113$, $\eta_p^2 = 0.03$, 95% CI $= (0.00, 0.13)$; Fig. 3B, right panels), as did the static asymmetric model $Q\text{-}asymm$ ($F(1,81) = 3.60$, $p = 0.061$, $\eta_p^2 = 0.04$, 95% CI $= (0.00, 0.15)$; Fig. S7B, central panels). In contrast, $Q\text{-}adapt$'s TI performance was differentially impacted by the change in underlying ground truth rankings, exhibiting a significant changepoint x direction interaction effect on non-anchor TI accuracy ($F(1,81) = 4.17$, $p = 0.044$, $\eta_p^2 = 0.05$, 95% CI $= (0.00, 0.16)$). This was driven by a significant improvement in TI accuracy from pre- to post-changepoint for 'up' participants modelled by $Q\text{-}adapt$ (pre-changepoint: mean accuracy $= 0.67 \pm 0.02$ SE; post-changepoint: mean accuracy $= 0.75 \pm 0.03$ SE; $t(38) = 6.56$, $p < 0.001$, $d = 1.05$, 95% CI $= (0.06, 0.11)$), in contrast to a far less pronounced, albeit still significant, increase in TI accuracy between changepoints for 'down' participants modelled by $Q\text{-}adapt$ (pre-changepoint: mean accuracy $= 0.68 \pm 0.02$ SE; post-changepoint: mean accuracy $= 0.72 \pm 0.02$ SE; $t(43) = 2.43$, $p = 0.039$, $d = 0.37$, 95% CI $= (0.01, 0.07)$). That is, $Q\text{-}adapt$'s non-anchor TI accuracy was relatively stunted in the 'down' group after the changepoint, whereas performance continued to improve in the 'up' group (Fig. S7B, right panels). Interestingly, the exact pattern of TI disruption in 'down' participants deviated from that predicted by $Q\text{-}adapt$; while our models predicted a more pronounced decline in lower-valued comparisons, the detrimental impact of the 'down' changepoint tended to be more strongly reflected in higher-valued comparisons (Fig. 2, lower panels). Given the similarity in effect sizes for the interaction effects under our two winning models $Q\text{-}asymm^2$ and $Q\text{-}adapt$, we next sought to more formally assess this apparent discrepancy in these models' ability to reproduce the key interaction effect of interest. To this end, we included 'model' as an additional within-subjects factor in the mixed ANOVA described above. This revealed a non-significant model x changepoint x direction interaction effect ($F(1,81) = 0.22$, $p = 0.637$, $\eta_p^2 = 0.003$, 95% CI $= (0.00, 0.06)$) – that is, the models did not significantly differ from one another in the extent to which they exhibited the direction-dependent effect of changepoint on TI accuracy.

Thus, evaluating not only the model evidence for each of our candidate models (i.e. 'predictive performance'), but also their ability to generate patterns of behaviour resembling those observed in humans (i.e. 'generative performance'[37]) revealed a somewhat mixed picture regarding the superiority of $Q\text{-}asymm^2$ versus $Q\text{-}adapt$ – while the former showed a clear advantage in terms of AIC, the latter was marginally better able to recapitulate the main behavioural effect of interest, albeit not significantly so. In either case, this lends support to the argument that human TI performance was best-captured by RL models whose belief-updating asymmetry does not remain static over the entire course of learning.

## Discussion

TI is an instance of humans' and other animals' impressive ability to utilise knowledge gained about local relations to infer global, unseen relationships. By introducing different changes in relational structure, we demonstrated that winner-biased belief-updating confers different levels of flexibility to adapt to such changes in ground truth orderings: whereas relocating the worst item 'up' to the top of the hierarchy is readily accommodated, relocating the best item 'down' to the bottom has a more disruptive impact on downstream inferential knowledge.

Participants' reduction in sensitivity to pre-changepoint TI comparisons with increasing combined value replicates compression effects previously observed in inferential learning settings[5]. Besides further underscoring the utility of using an RL framework to capture TI learning dynamics[4,5,38], we extend these findings by observing differences in adaptability to changes in relational structure that are consistent with an asymmetric, rather than symmetric, update rule. Our findings lend further credence to the hypothesis that belief-updating asymmetries extend beyond two-armed bandit and foraging task contexts[10]. We note that the specific form of positivity bias in the present study is somewhat different to those investigated in the wider literature. In other RL paradigms, 'positivity' refers to the preferential update of values upon receipt of a positive (as opposed to negative) reward prediction error. Here, in contrast, the bias lies in the disproportionate updating of the winner and loser of a given binary comparison, independent of the sign of the prediction error.

Our paradigm's minimal change in underlying ground truth structure halfway through the task was reflected in a slight change in feedback that only subtly differed between groups: both sets of participants were given a single new piece of feedback (i.e. $i_7 < i_1$), and only differed in the single comparison pair that no longer offered feedback (i.e. $i_1$ vs. $i_2$ for 'up' participants, and $i_6$ vs. $i_7$ for 'down' participants). To model the updating of item value estimates in response to choice feedback, we assumed a relatively simple RL framework that only updated its cached value estimates for the currently presented pairs of items on receipt of feedback. Indeed, the utility of this 'model-free' approach in capturing human TI behaviour demonstrates that such inferential capabilities can proceed without necessarily invoking any abstract knowledge of the structural regularities entailed by particular relations (i.e., knowing that A < C *because* A < B and B < C). Nonetheless, it remains an intriguing possibility that humans could have resolved the ambiguity initially induced by the changepoint by learning from trials from which they receive no feedback. In the present case, for example, a participant in the 'up' group might have learned to expect feedback, given the presentation of $i_1$ vs. $i_2$. The subsequent, unexpected omission of this feedback after the changepoint could induce them to update their value estimates for the presently compared items, and/or indeed items at the other end of the hierarchy, since it could be seen as diagnostic as to which of the underlying changes in ground truth explains the recently observed and highly surprising outcome $i_7 < i_1$. This capacity to infer how the receipt or omission of feedback on a given comparison bears on items elsewhere in the hierarchy would therefore require a model of how the full set of items are related, potentially drawing on model-based approaches furnished with the ability to mentally simulate the outcomes of pairwise comparisons through replay[39–41].

The compression of participants' learned value structures constitutes an instance of a more generalised distortion widely observed across psychophysical, numerical, and economic decision-making contexts, whereby the discriminability between comparanda decreases with increasing stimulus intensity or magnitude[42–46]. Here, we propose that such compressed

representations may emerge from an asymmetric learning rule (see also ref. 5). Nonetheless, we by no means argue that belief-updating biases are the only source of these ubiquitously observed psychometric distortions. We note that the reduction in discriminability owing to increased overall value estimates across the hierarchy is not inconsistent with the view that compressed judgements of magnitude may arise, for example, from the mental organisation of numerical information on a power or logarithmic scale[44,45]. Indeed, we observed a divergence in how human and simulated TI performance was impacted by the 'down' changepoint as a function of combined pair value, with higher valued comparisons appearing to be most strongly disrupted in humans, but less so in our simulated asymmetric agents. This discrepancy may imply the presence of an additional compressive force that further reduces the discriminability of higher-valued comparisons. One potential way of disentangling the relative contributions of asymmetric policies and non-linear 'Weber scaling' of internal representations in the relational learning domain would be to more closely examine participants' individual differences in the sign of the asymmetric learning bias: if the behavioural compression effects exhibited by winner-biased participants were equivalent to the anti-compression effects of loser-biased participants with equal *absolute* learning rate asymmetries, then this would further emphasise the role of asymmetric learning policies in the emergence of value compression. Relatedly, observing the opposite changepoint x direction interaction effect observed in our experiment, but among a predominantly *loser-biased* population – that is, disruption to inferential knowledge among the 'up' group, rather than the 'down' group – would lend further support to the idea that it is the sign of the learning asymmetry that is responsible for any (in-)efficient changepoint adaptation effects. Given the limited number of loser-biased participants in the present study, we leave this question for future work containing larger and more diverse samples of participants.

Our behavioural predictions were derived from *Q-asymm*, an RL agent that scaled its updates of winners and losers of pairwise comparisons according to asymmetric learning rates that remain fixed throughout the experiment. While this model significantly outperformed its symmetric counterpart *Q-symm*, it nonetheless underestimated the proportion of winner-biased participants. This raised the possibility that well-performing participants in the 'down' group were capable of both adapting to the change in relational structure, while also exhibiting pre-changepoint compression effects consistent with initially winner-biased learning. We therefore introduced two additional models with differing degrees of modulation in their learning asymmetry. *Q-asymm²*, a model that undergoes a one-shot reset in learning rates after the changepoint, convincingly exhibited the strongest predictive performance, but is agnostic about how such changepoint-induced changes in learning asymmetry may arise. In contrast, *Q-adapt* modulates its trial-by-trial learning rate asymmetry as a function of the strength of its choice preference. Theoretical and empirical work alike has indicated an essential role of uncertainty in shaping how animals do, or indeed ought to, learn from reward feedback, guiding the use of habitual versus goal-directed control in humans[47], the flexible combination of reward information in primates[48], and the volatility-induced adaptation of learning rates via meta-learning in rodents[49]. Indeed, recent TI work has suggested that the uncertainty of pairwise comparisons may serve to earmark them in a way that makes them more labile for subsequent structural reorganisation[18]. In contrast, in modulating its learning asymmetry as a function of its preference for one of the two compared items, *Q-adapt* proposes a potential mechanism for how this asymmetry modulation may arise, while also unifying recent findings that asymmetric and symmetric learning policies respectively best explain human behaviour in, and indeed are optimal for, partial and full feedback TI regimes[5]. *Q-asymm²* exhibited a significant improvement in model fit relative to both the static learning model *Q-asymm* and the adaptive model *Q-adapt*. We tentatively note that only *Q-adapt* quantitatively reproduced the significant direction-dependent impact of the changepoint on downstream inferential accuracy, although formal comparison between these interaction effects under the two winning models yielded no statistically significant differences. In either case, our

findings indicate that humans can adapt their predominantly winner-biased learning asymmetries, raising questions about if and how this adaptability is shaped by uncertainty. One promising avenue for future exploration is to consider how dissociable forms of uncertainty may differentially influence this learning asymmetry. Existing models of changepoint adaptation typically possess the ability to separate the 'aleatoric' uncertainty pertaining to expected variability in an outcome from the 'epistemic' uncertainty arising from unexpected changes in a volatile reward environment[12–14]. Changepoints cause these models to increase their learning rates until the period of high epistemic uncertainty is resolved. While previous work has found evidence for asymmetric TI learning in stochastic feedback settings[5], our task involved deterministic feedback, making the changepoint-indicative feedback relatively salient. It therefore remains an interesting open question whether humans' winner-biased belief-updating asymmetries, and their ability to adaptively change them, intersect with their ability to disentangle objective relational changes (i.e., volatility) from noisy choice feedback (i.e., stochasticity).

Several lines of research have connected elementary belief-updating biases with research in clinical settings. While positivity biases may provide an adaptive means of promoting positive well-being[50] or motivation[6], converging empirical and theoretical work has also implicated more pessimistic learning rates in a range of symptoms of Major Depressive Disorder[51–53]. Our finding that belief-updating biases confer different levels of flexibility to changes in relational structure raises interesting questions about whether or not such differences in adaptability also cut across clinical populations. It would be particularly interesting to consider such asymmetries in changepoint adaptability, particularly during relational learning, in the context of risk-seeking or gambling behaviour, since they predict differences in the influence of various outcomes – e.g., a change in a previously low-performing bet versus a change in a previously high-performing bet – on reward expectations pertaining to unchanged bets. While our paradigm contained fully deterministic relational feedback and therefore did not incorporate any risk or outcome variance per se, evidence suggests that the degree of learning asymmetry shapes the relationship between the environment's reward variability and an individual's tendency to seek or avoid risks[54–56]. It would therefore be worthwhile to consider the role of belief-updating biases in value compression and changepoint adaptability effects under different levels of outcome variance (i.e., via probabilistic relational feedback), and how this relationship might be moderated by clinically relevant symptoms or traits.

Our RL agents constituted descriptive models of how biased learning policies give rise to subjective value distortions and differences in behavioural adaptability. While we make no causal or mechanistic claims about the dynamics of relational learning in the brain, research centring on neuromodulatory activity in the basal ganglia and brainstem may offer plausible accounts for how updates may be asymmetrically scaled during RL. More specifically, subpopulations of striatal neurons with distinct excitatory and inhibitory properties (i.e., D1 and D2 receptors, respectively) may provide a means of differential engagement of dopamine-mediated learning as a function of positive or negative prediction errors[57–59]. Likewise, empirical work has implicated serotonergic systems operating over behaviourally relevant timescales in the ability to track and adapt to changes in the volatility of reward environments[49,60]. It would therefore be interesting to consider how such neural accounts extend beyond bandit tasks to structure learning settings that more explicitly engage generalisation and inference processes. In addition, our investigation of differences in adaptability to changes in underlying relational structure ties into research exploring how neural and artificial systems reconfigure knowledge at the representational level in response to new information. Evidence suggests that the linking together of transitive hierarchies is mirrored in the joining of neural manifolds in fronto-parietal regions and deep neural networks alike[18]. Examining how the differences in relational adaptability observed in the present study might be recapitulated in a neural network may, in turn, yield neuroscientific hypotheses about how representational geometries may be

(in-)efficiently reorganised in response to changes in environmental structure.

## Limitations

Since our primary focus was on how relational changes impact structural knowledge, we sought to identify participants who had learned the transitive hierarchy to a sufficient degree before the changepoint onset. This entailed applying the same performance threshold used by Ciranka et al.[5], but focused on pre-changepoint trials. This amounted to a somewhat conservative approach, with a relatively high proportion of participants being excluded (see 'Participants' in *Methods*). While focusing our analyses on participants who had already acquired adequate inferential knowledge was necessary for probing any changepoint-induced changes in this knowledge, it does raise the question of the generalisability of our claims about how learning asymmetries confer different levels of adaptability to relational changes. Nonetheless, we note that applying a more liberal threshold for inclusion ($\alpha = 0.1$, rather than 0.01), which resulted in $N = 103$ participants ('up': $N = 53$; 'down': $N = 50$), left our core findings unchanged. That is, we likewise observed a differential impact of the changepoint on downstream TI performance that was best captured by $Q\text{-}asymm^2$ (Fig. S8 and Fig. S9A, B).

We also acknowledge that the RL models implemented in the present study constitute only a narrow subset of the ways in which we can understand inferential learning and the cognitive processes that support it. For instance, one could alternatively examine the TI changepoint problem under a Bayesian inference scheme, as has widely been done in the context of TI[3,38,61], and indeed structure learning more generally[62–64]. Under this broad class of frameworks, our TI changepoint scenario could be viewed as a problem of resolving feedback ambiguity: the new observation that $i_7 < i_1$ is, at first, equally consistent with a change in $i_1$'s ranking as it is with a change in $i_7$'s ranking, meaning the agent must track the likelihood of subsequent choice feedback under each of these hypotheses about the new underlying ground truth structure. Whereas our models maintain only a point estimate of each item's value, sampling algorithms that maintain a posterior distribution over item estimates (e.g., ref. [38]) may be better suited to probing how uncertainty about items occupying different portions of value space might evolve over time. A wealth of literature has likewise researched the role of episodic processes, likely implemented in the hippocampus, in enabling inference and generalisation[65,66]. More specifically, TI may be supported by 'retrieval-based' inference mechanisms that reactivate and recombine pattern-separated representations of specific relations[41,67], or via a more 'encoding-based' recruitment of inferred relationships via overlapping structural representations[68,69]. Indeed, we note that Ciranka et al.[5] found that incorporating an additional 'pair-level' memory component, which allowed for episodic retrieval of pairs for which feedback was delivered, improved model fit by capturing enhanced learning for direct comparisons between neighbouring items. Since the present study focused on how the reorganisation of relational knowledge intersects with widely observed biases in value learning, we did not incorporate into our models the possibility that transitive learning might also involve episodic memory processes (cf. refs. [5,41]). Elucidating whether and how such episodic processes feed into the caching of item values would therefore be a promising avenue for future work.

## Data availability

The numerical data that support the findings of this study are publicly available in .csv format at https://doi.org/10.6084/m9.figshare.26147470.

## Code availability

The experiment and analysis code can be found at https://doi.org/10.5281/zenodo.17257543.

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

## Acknowledgements

We are grateful to Peter Dayan for valuable discussions and for comments on the manuscript. We also thank Christian Doeller for helpful feedback

during project conceptualisation, and Philip Jakob and Maik Messerschmidt for technical support. This work was supported by Deutsche Forschungsgemeinschaft (DFG) Research Grants (DFG-SP-1510/6-1) and (DFG-SP-1510/7-1), and a European Research Council Consolidator Grant (ERC-2020-COG-101000972) to B.S. T.A.G. is supported by the Max Planck School of Cognition. The funders had no role in study design, data collection and analysis, decision to publish, or preparation of the manuscript.

## Author contributions

The following list of author contributions is based on the CRediT taxonomy: Conceptualisation: T.A.G., B.S. Data Curation: T.A.G. Formal Analysis: T.A.G. Funding Acquisition: B.S. Investigation: T.A.G. Methodology: T.A.G., B.S. Project Administration: T.A.G., B.S. Resources: B.S. Software: T.A.G. Supervision: B.S. Validation: T.A.G. Visualisation: T.A.G. Writing - Original Draft: T.A.G. Writing - Review & Editing: T.A.G., B.S.

## Funding

## Competing interests

The authors declare no competing interests.
