## [Transparent Peer Review file · Communications Psychology]

Asymmetric learning and adaptability to changes in relational structure during transitive inference

Corresponding Author: Mr Thomas Graham

Version 0:

Decision Letter:

Dear Mr Graham,

Thank you for your patience during the peer-review process. Your manuscript titled "Asymmetric learning and adaptability to changes in relational structure during transitive inference" has now been seen by 3 reviewers, and I include their comments at the end of this message. They find your work of interest but raised some important points. We are interested in the possibility of publishing your study in Communications Psychology, but would like to consider your responses to these concerns and assess a revised manuscript before we make a final decision on publication.

We therefore invite you to revise and resubmit your manuscript, along with a point-by-point response to the reviewers. Please highlight all changes in the manuscript text file.

Editorially, we consider it crucial that the revised manuscript includes additional model comparisons that are necessary for strengthening the conclusions regarding asymmetric learning and adaptability, as requested by reviewer 1 and 2. Furthermore, the clarifications of the methodological and theoretical concerns, such as the instruction of the change in "narcissism" and the large exclusion rate, are also important in the revised version.

I am attaching an Editorial Requests Table that details critical reporting requirements for the revised manuscript. Please attend to each item and ensure your manuscript is fully compliant. We are requesting that your manuscript aligns with these requirements as this facilitates the evaluation of your manuscript, reducing delays in re-review and potential future acceptance. If your revised manuscript is not aligned with these requests on major issues, such as those concerning statistics, it may be returned to you for further revisions without re-review. Additional information can be found in our style and formatting guide <https://www.nature.com/documents/commspsychol-style-formatting-guide-accept.pdf> Communications Psychology formatting guide.

Please use the following link to submit your

- revised manuscript,
- point-by-point response to the referees' comments,
- cover letter (as a separate document),
- the Editorial Policy Checklist (see below),
- the Reporting Summary (see below), and
- the completed Editorial Request Table (attached):

Link Redacted

We hope to receive your revised paper within 8 weeks; please let us know if you aren't able to submit it within this time so

that we can discuss how best to proceed. If we don't hear from you, and the revision process takes significantly longer, we may close your file. In this event, we will still be happy to reconsider your paper at a later date, provided it still presents a significant contribution to the literature at that stage.

Best regards,

Troby Lui

Troby Lui, PhD
Associate Editor
Communications Psychology

REVIEWER EXPERTISE:

Reviewer #1: transitive inference, reinforcement learning

Reviewer #2: decision making, reinforcement learning

Reviewer #3: transitive inference, reinforcement learning

REVIEWER REPORTS:

Reviewer #1 (Remarks to the Author):

Summary: This paper examines how people adapt to changes in relational organization through incremental learning from local comparisons. The authors use a transitive inference (TI) task in which participants first learn a relational hierarchy of 7 items, but halfway through the task there is a sudden change such that either the lowest- or highest-ranked item moves to the other end of the hierarchy. Following on earlier work showing that people demonstrate asymmetric belief updating in a similar type of task, here the authors examine whether such asymmetric biases in learning help or hinder the ability to adapt to a sudden change of this sort. Using relatively simple model-free reinforcement learning models, they show first through simulation that a winner-based learning policy of the kind seen in previous work would lead to successful learning of a positive change in rank (i.e., where the lowest-ranked item becomes the highest-ranked) but comparatively worse learning of a negative change in rank (i.e., where the highest-ranked item becomes the lowest-ranked). Results from the behavioral experiment show that 1) participants largely exhibit a winner bias in belief updating, replicating a previous study, and 2) after the change point, participants readily adapt to the upward change in rank, but show impaired learning after the downward change in rank. The RL modeling indicates that models with asymmetric value updating (Q-asymm) generally outperform symmetric value updating (Q-symm) in fitting these data. Further exploration of the results indicated that there were some participants who initially exhibited strong biases toward winner-based updating, yet who readily adapted to downward shifts in rank. The authors present a further model (Q-adapt) that adaptively changes the degree of bias depending on the strength of the preference between two items, where violations of strong predictions lead to symmetric updates, but asymmetric updating may still apply for comparisons with more uncertainty about relative value.

This paper has several notable strengths. It is generally well-organized and well-written, with a clear rationale and in-depth coverage of relevant past work. The methods and analyses appear to be appropriate for the research question and hypotheses. The authors have provided an open repository that includes the raw data, experiment files, and analysis files, and seem reasonably well-documented to support reproducibility.

The core behavioral findings are interesting and constitute a meaningful contribution to this research area. The authors replicate their previous finding of a winner-bias in belief updating in this task, helping to establish the robustness of that effect. The more novel finding is that, despite relatively minimal changes in feedback after the change point, people clearly seem less able to adapt to downward shifts in rank, including their knowledge of items whose ranks didn't change. The authors propose a reasonable mechanistic theory that might explain this phenomenon and which meaningfully builds upon the idea of an updating asymmetry.

There are a couple of weaknesses that stood out to me. The first concerns the large amount of data that was excluded (see point 1 below). The other broad concern is that, while the Q-adapt model is an interesting proposal, the evidence for it seems somewhat weak on the whole given the current data. The model recovery simulations don't imply "robust recoverability" to me, as there's a fair amount of confusion between the Q-asymm and Q-adapt models. The Q-adapt model also did not outperform the Q-asymm model on one of the criteria (comparisons of AIC scores). It's also unclear whether there's evidence for the trial-by-trial adaptation of learning rates proposed by the authors vs. a more discrete difference between the two phases (see point 4 below). In general, the Q-adapt model seemed to introduce a great deal of further complexity on top of the Q-asymm model, leaving it a bit unclear what aspects were essential for capturing the behavioral effect. That being said,

while it doesn't seem all that parsimonious to me, I think the authors nicely explain the rationale for the proposed model and how it connects to related ideas in RL, and in that sense the theoretical contribution may outweigh the somewhat tentative nature of the evidence for the proposed mechanism.

Main points

1. Nearly half of the participants were excluded based on their performance prior to the change-point. I understand the reason for focusing only on people who demonstrate some learning of the hierarchy, but such a large proportion of excluded data is a bit surprising and may speak to the relative difficulty of the task (or perhaps broader concerns about data quality with an online sample). The ramifications of this for the generalizability of the conclusions are unclear to me (e.g., perhaps the observed winner asymmetry is overestimated because it was less common among the lower-performing participants who were excluded). I think the authors should minimally acknowledge the rate of exclusions more directly in the main text of the paper, and consider discussing the extent to which this may impact their conclusions. It may also be helpful to report other signals of data quality that could corroborate exclusions if they are available.
2. The authors highlight the intriguing result that a downward shift in rank seems to impair TI performance for other, non-anchor items (items 2-6) despite nothing changing about the feedback received for those items. I struggled a bit to understand how this arises in the model, and I think the authors could clarify the effects on "downstream inferential knowledge" (e.g., by moving the explanation in lines 174-178 to somewhere earlier in the paper). There are some early claims about these effects which are somewhat vague and make it difficult to appreciate that it's not only the slower downward adjustment of item 7, but also the fast upward adjustment of item 1 that together produce worse discriminability for lower-ranked items overall.
3. The authors report a basic analysis indicating that participants adjusted equally well to the upward and downward shifts in terms of comparisons involving the anchor items (1 or 7). It wasn't clear to me why the change in the anchor items wasn't included in Figure 2 showing the changes over time, since the models also make different predictions about the speed with which estimates of those items change. Although the overall comparison wasn't significant, the means at least suggest a slower adjustment of item 7 for the downward shift compared to item 1 in the upward shift. In any case it seems an important feature of the data to provide alongside the changes in accuracy for non-anchor items (and which I found hard to grasp from the matrices in Figure 3).
4. As I noted above, the Q-adapt model involves a big jump in complexity, which left me wondering how well it could be validated with the current data. The authors also fit a simpler model (described at bottom of pg. 13) with separate learning rates for each phase (pre- and post-changepoint), but this is not included in the model comparisons. This model could be seen as a simpler version of the "adaptive asymmetry" explanation: That participants adjust their learning rates once after receiving feedback that strongly violates existing beliefs. I'd be interested in how this compares to the other models, and more generally whether there's any other evidence for the trial-by-trial adjustment of learning rates based on the confidence for the current pair.

Reviewer #2 (Remarks to the Author):

Graham and Spitzer report a study that aims to demonstrate that when people update relational representations, they adapt the degree to which they use an asymmetric learning possibility. They found that participants easily learned that a previously low-ranked item now had a higher rank. However, participants had a harder time learning that a previously high-ranked item now had a lower rank. Computational modeling suggested that participants exhibited a winner-biased learning policy, but adapted this strategy when they needed to learn a new relationship with an new inferior option.

This is an interesting paper, with a sophisticated computational modeling technique and convincing data. The paper is not squarely in my field of expertise (which is in decision making and computational modeling, but more on the side of cognitive control), so my comments may come a bit out of left field (which may be good given the broader readership of this journal). I don't have many comments about the design of the experiment, and the analysis of the data. These appear quite solid.

My major concern, however, is that the scope of this paper appears relatively narrow. From my (outsider) understanding, it was already clear that people use a winner-biased updating rule. If this is the case, then it should be relatively obvious that people are going to perform better in the "up" compared to the "down" condition. The only remaining novelty then is that people seem to adapt their learning policy based on the context. I am worried that this may not be a big enough advancement in understanding in the field to warrant publication, but I'll readily admit that I am not in the best position to comment on this.

I think Figure 2 can be improved. First, it is not immediately obvious what the different colors represent. There is a little matrix on the right that also contains these colors, and some labels ("low", "medium", "high") but one needs to read the caption to understand that this is the true difference in rank. More importantly, however, the moving window analysis induces apparent changes in behavior before the onset of the changepoint. For example, if we look at human behavior in anticipation of the change point, it looks like humans already start adjusting the value of the items (especially in the "down" condition where they seem to flatten before the changepoint in comparison to the "up" condition). I recommend that the authors find another way of plotting the data, so that post-changepoint adaptations do not bleed in the pre-changepoint behavior.

The paper never makes it clear what “cnarcy” means. I surmised that this is a made-up word, but the authors never explicitly state this. I think the purpose of using this word deserves a little bit more attention. I am also not quite sure why the authors chose this non-word, because (at list in my mind) it sounds like “snarky”, which is a real English word. I understand that the study was conducted in Germany, but many young Germans are quite comfortable with English (especially in higher education), and so I am afraid that they may use this definition of the word to make their initial judgments. This would make their initial rank order not so neutral. Of course, this concern is present in both the up and down conditions, so its effect should cancel out in the long run. Still, this potential interpretation of the word stood out to me.

It was not clear to me from the text how “adjacent and TI trials were randomly interleaved within each of six blocks”. Can the authors add more detail?

When the authors inspect the development of TI accuracy after the changepoint, they divided this phase in half and used the “half” as a within-subjects factor. This was surprising to me, because trial number is a nice continuous measure that already represents an increase in time. Why did the authors make this predictor categorical? It seems like a lot of useful variance is thrown away by doing this. For example, they could just add trial number as a continuous regressor in an LME model.

It seemed arbitrary to me that the learning rate for the Q-adapt model can't reverse. The authors cite some literature suggesting that this does not happen, but the task certainly invites a reversal (and it seems optimal as well). Why not fit a model that allows for reversals to happen, and investigate whether relieving this restriction provides a better model fit.

Reviewer #3 (Remarks to the Author):

Graham and Spitzer present an interesting and clearly laid-out study that investigates the role of asymmetric reinforcement learning in learning generalizable relations; in particular, in the naturalistically relevant case where subjects must learn changes in underlying relational structure. Relational learning and the transitive inference task paradigm they adopt are important areas in the study of cognition, and the RL-based models they focus on are an important model class in clarifying learning mechanisms in humans and animals. This study is an extension of recent innovative work from their group (Ciranka et al. 2022) that showed that asymmetric learning (i.e. here, in the transitive inference task paradigm, updating of model parameters preferentially for items that are either “winners” or “losers”) actually facilitates relational learning, and that such learning well-matches experimental behavioral data. In our opinion, the themes of this work will be of interest to researchers across different fields (relational learning, RL). We also highlight that the authors' aim to investigate cases of change to underlying relational structure represents a valuable extension to existing relational learning studies.

The authors extend their investigation of asymmetric learning more specifically to changes in relational structure by introducing a variant of the transitive inference task in which there is a shift in transitive (rank-based) structure, either where the highest- (or lowest-) ranking item becomes the lowest- (or highest-) ranking item. The authors collect data from human subjects on this task variant, from which they report a number of findings, mainly (1) human subjects generally perform better when a lowest-rank item is maximally promoted in rank (rather than when the highest-rank item is maximally demoted), in agreement with the basic expectation from their previous work, (2) across the task changepoint, a subset of human subjects appear to adapt their asymmetric learning from winner-based to loser-based, doing so in a manner correlated with task performance, and (3) an RL model that the authors introduce, which has the ability to adapt its asymmetric learning bias, provides a relatively better match (than a non-adaptive model) to the human data.

Overall, we found this study to be interesting and thorough, and we appreciate the clarity of the authors' written presentation. We do however have several points that we would like to see addressed (main requests), in addition to some additional requests for improving the manuscript.

Main requests

- We request some clarification regarding the explicit prompting of the subjects to expect a change in relational structure (in lines 678-681 in Methods, the authors write: “At the start of the experiment and before each block, participants were reminded that not all items would stay as cnarcy for the entire experiment. Rather, on some blocks, certain items may or (may not) become more or less cnarcy, meaning their relations to other items (as reflected in choice feedback) may change.”) While we have no problem in principle with this aspect of the task, we believe a few things are appropriate:

- First, we think the authors should clarify in the manuscript (ideally in both main text and Methods) what the subjects were informed of regarding the possible changes to ‘cnarciness’, i.e. whether it would change at any trial (or for an entire block) and whether the change was completely arbitrary (i.e. a single item or any set of items? any kind of specific change in rank?). It would probably be most helpful simply to have the actual prompt used directly reported to the Methods section (and also with some adequate clarification in the main text).

- Second, the authors should provide some additional discussion (either in the Methods or, even better, in the Discussion) of how this point might alter the asymmetry adaptability or other learning/decision aspects (e.g. winner-/loser- asymmetry bias, decision noise) of the human subjects. At the very least, it would be helpful to clarify where the authors stand regarding the expected generality of their results and some insight into their rationale.

- We think these are reasonable requests since in many (if not most) real-world scenarios (even the authors' example of teams winning and losing) there are clearly situations where there is not an explicitly prompted expectation of relational change. Further, it is reasonable to expect that such prompting could promote asymmetry-based adaptability. We do not think such a possible effect makes the findings less interesting, but would better delineate the impact of the findings in the paper if addressed openly.

- The authors find an interesting behavioral pattern in the human subjects in the “down” group, which remains unaccounted for by their RL models, namely the decrement in performance in the high trials after the changepoint (seen in Figs 2 and 3). This seems to be quite a striking (interesting) experimental result, but apart from noting its existence, it is not discussed further by the authors. It seems some discussion is merited, perhaps ideally in the Discussion section, with regard to why the human subjects show this, and/or possible modifications to the Q_adapt model (or other model).
- It also seems appropriate to include some discussion (again, perhaps solely in Methods, but certainly possibly in the Discussion) regarding whether and how the use of explicit TI (where the transitivity- or rank-based relation is explicitly told to the human subject, “cnarciness”) would have an impact on symmetric vs. asymmetric learning biases, and on adaptability as well (this interacts with our request above regarding the explicit prompting of possible relational change). Given that -both- the transitive relation and the possible reconfiguration of the transitive hierarchy are explicitly prompted in the task the authors use, it seems particularly appropriate to speak to this issue directly. Further, as work on TI in animals is basically implicit, providing some discussion would help guide thinking on possible connections to the wider literature on TI. For instance, though implicit TI tasks (i.e. where the subjects are merely tasked with choosing between stimuli, and are not prompted about ordering (e.g. “cnarciness”)) are almost certainly more difficult overall, it seems plausible to us that subjects who do perform implicit TI may show asymmetric learning (and, when controlling for performance, perhaps even more asymmetric learning than in explicit TI). (Given that TI in human subjects has been found to occur successfully even when subjects are not consciously aware (e.g. Greene Spellman et al. 2001), something like winner-/loser- bias and adaptability seem plausible: perhaps subjects do not need to be consciously aware or semantically know about the task in order for them to exhibit asymmetric learning/adaptability). If this scenario or some other is reasonable to postulate, then we think it would be beneficial to various readers to have the authors weigh in.

Additional requests

- In the abstract / introduction, the term “choice preference strength” is used — however, we think that for first-time readers, this term/concept will not at all be clear. To address this, can the authors unpack the phrase or use an alternative phrase (it is fine also to retain this phrase, but in a parenthetical).
- line 602-607: regarding positive or negative RPE — we suggest providing some further brief clarification of relationship (either here or earlier in manuscript) between RPE and winner/loser-bias. Currently the authors simply say “plausible accounts”, but this is too vague to be useful.
- While the connection is indirect, there is some interesting degree of conceptual similarity between the present study’s choice-preference (or “belief”) based modulation (of asymmetry learning) and the certainty-based updating mechanism for transitive knowledge assembly in the Nelli et al. 2023 study. Perhaps it is worth incorporating some discussion of this parallel, and even possible neural-level basis or substrates (if any), in the Discussion section.
- We think it may be appropriate to provide (in a supplement) informal presentation of some of the debriefing responses, if they shed any light on the asymmetry bias and/or on the adaptive asymmetry modulation. Of particular interest could be subjects that show the extreme shift from winner- to loser-based asymmetry according to the Q_asymm2 model (Fig. 4D). Though informal, this could help provide some intuition (affirming/disaffirming) whether and to what extent the learning biases were consciously accessible and deliberately executed.
- In the authors’ previous study on asymmetric learning (Ciranka et al. 2022), they find that the RL model best matching experimental data (the model is termed “Q2* + P”) incorporated some degree of direct pair learning. If only in the Methods section, can the authors provide some clarification why this model was not adopted in the present study?
- For Fig. 3., it appears to be appropriate to provide results from Q_asymm in this figure (in between Human and Q_adapt), if simply to provide consistency with other figures showing both Q_asymm and Q_adapt, unless there is a clear reason not to.
- For Fig. 3B, the authors’ choice of a monotonic color scheme is inappropriate. To clarify differences in sign (and not just magnitude), they should use a diverging color scheme (such as red-white-blue, where red is negative values and 0 is white, and blue is positive values; or some other similar color scheme).
- Line 300: There should be a reference to Fig. 4C.
- Fig. 4D: Larger binsizes (perhaps twice as large) seem appropriate.
- Lines 327-339 (and Fig. S4). Perhaps a Spearman correlation is more appropriate than a Pearson correlation (or complementary). Further, it would be helpful to include the correlation statistics in Fig. S4 caption instead of just in main text.
- Lines 369-377: The introduction of lambda is a bit confusing and unintuitive on first read. To address this, we recommend two changes: first, by changing the parenthetical “(indicating a weak preference)” to “(corresponding a weak choice preference, or weak “belief”)”, and similarly for “indicating a stronger preference”; second, by changing “reflects” (line 370) to “is dependent on”.
- line 385: “strong prior belief”. Here and elsewhere, is the word “belief” strictly warranted as equivalent to the strength of choice preference? (Even though likely true in many subjects doing the task.) Perhaps the use of scare quotes for the word “belief”, and/or perhaps the use of a phrase or description closer to “choice preference strength” is safer.
- line 307: “leading to inflated estimates of loser learning rates.” This is somewhat confusing, as it seems like the model could also lead to deflated estimates of loser learning rates. Perhaps there is a better way to state this.
- p. 404: “choice parameterizations” : It would be helpful for the authors to clarify what this phrase refers to, using a parenthetical.
- p. 439-446 - In Fig. 3B, the authors should provide plots of change in TI accuracy instead of pre-post change in choice preference (or do so in addition to pre-post change in choice preference in an added subpanel). Not only because this is more directly what the main text is referring to (in line 435), and also because this would serve to clarify the change in behavior across the changepoint.
- Fig. 5B: The caption should clarify what ppx estimated frequency is.
- line 517: Instead of “once the changepoint was reached”, we would recommend “by the time the changepoint was reached.”

- line 558: "relative balance..." — it's a bit odd to talk about "narrowing" a "balance" (these words don't naturally go together) — perhaps consider "balance of positive and negative learning rates may dynamically become more even" OR "difference between positive and negative rates may dynamically be eliminated, rather than change in sign"
- line 602: consider adding "...behavioural adaptability in a relational setting" or "behavioural adaptability in the setting of a relational task"
- Methods (638-651): can you provide some direct reportage of overall performance in included groups at both criterion levels (0.01 and 0.1)?
- line 839 — "Fig S4C" is cited but there is no included S4C panel in the supplementary file. If the authors meant to include some (at least examples) of model parameters that may be traded-off but were not, that would be helpful.

EDITORIAL POLICIES

We ask that you ensure your manuscript complies with our editorial policies and reporting requirements.

To that end, we require revised manuscripts to be accompanied by two completed items: a reporting summary that collects information on study design and procedure, and an editorial policy checklist that verifies compliance with all required editorial policies.

- <https://www.nature.com/documents/nr-reporting-summary.zip>>Nature Research Reporting Summary
- <https://www.nature.com/documents/nr-editorial-policy-checklist.pdf>>Editorial Policy Checklist

All points on the policy checklist must be addressed. Your revised manuscript can only be sent back to the referees if these checklists are completed and uploaded with the revision.

Notes: If you have submitted a Stage 1 Registered Report, Review, Primer, Comment, or Perspective you do not need to submit these forms. If you have already submitted these forms, you may disregard this request.

** Visit Nature Research's author and referees' website at <http://www.nature.com/authors>>www.nature.com/authors for information about policies, services and author benefits**

If you experience problems in linking your ORCID, please contact the <http://platformsupport.nature.com/>>Platform Support Helpdesk.

Version 1:

Decision Letter:

Dear Mr Graham,

Thank you for your patience during the peer-review process. Your manuscript titled "Asymmetric learning and adaptability to changes in relational structure during transitive inference" has now been seen by 3 reviewers, and I include their comments at the end of this message. They find your work of interest but raised some important points. We are interested in the possibility of publishing your study in Communications Psychology, but would like to consider your responses to these concerns and assess a revised manuscript before we make a final decision on publication.

We therefore invite you to revise and resubmit your manuscript, along with a point-by-point response to the reviewers. Please highlight all changes in the manuscript text file.

Editorially, we consider it crucial that Reviewer #2's concern regarding the Q-adapt vs. Q-assym² prediction on the changepoint*direction interaction is thoroughly addressed.

I am attaching an Editorial Requests Table that details critical reporting requirements for the revised manuscript. Please attend to each item and ensure your manuscript is fully compliant. If your revised manuscript is not aligned with these requests on major issues, such as those concerning statistics, it may be returned to you for further revisions without re-review.

Please submit the following items:

- Revised manuscript
- Point-by-point response to the referees' comments
- Cover letter (as a separate document)
- <https://www.nature.com/documents/nr-reporting-summary.pdf>>Nature Research Reporting Summary
- Completed Editorial Request Table (attached).

via this link: Link Redacted .

Additional guidance is available in our style and formatting guide <https://www.nature.com/documents/commpsychol-style-formatting-guide-accept.pdf>>Communications Psychology formatting guide.

Best regards,

Troy Lui

Troy Lui, PhD
Associate Editor
Communications Psychology

REVIEWER REPORTS:

Reviewer #2 (Remarks to the Author):

In this revision, Graham and Spitzer have thoroughly responded to my comments. I appreciate the new continuous mixed effects model, as well as the new Q-assym² model that allows for the order between positive and negative learning rates to be different pre- and post-change point. I think the paper is strong, and I am looking forward to recommending acceptance when the following remaining concerns are addressed.

It's good to see that this model provides a better fit to the data than Q-adapt. However, I am not particularly convinced by the set of simulations in which the authors show that the Q-adapt, but not Q-assym², predicts the change point * direction interaction observed in human behavior (where performance post change point increases for the 'up' condition but not for the 'down' condition). At the core of this criticism lies the observation that the simulations from both models seem to show this effect (see Figure 2 rightmost panels). Moreover, the stats of the sets of ANOVAs (effect sizes and p-values) are close, which makes me wonder if the two models truly make such different predictions. First, you want to show that the pre-post by interaction effect in the q-adapt data is stronger than in the q-assym² data (see Nieuwenhuis et al. 2011 for a formal description of this problem). Most importantly, however, it seems odd to base this conclusion on just one set of simulations. I understand the authors simulated one dataset for each model so they could perform the same set of analyses as on the behavioral data. However, the noise from these simulations makes it hard to determine whether the authors have truly discovered that the fits from q-assym² do not make this prediction, or whether they just found a dataset in which the interaction did not occur by random chance. (This concern is exacerbated by the relatively similar stats of the two ANOVAs.) A more convincing version of this analysis would be to run a large number of simulations (say 100 or 1000 sims), and then count how many times each model includes a significant interaction effect. That would rule out any uncertainty about random chance influencing this validation check of the data.

I think it would also be good to mention that 'cnarciness' was a deliberately chosen nonword in the main text (where it is first encountered), and not just in the methods.

Finally, I found new sentence in the results section a bit awkward:

Participants were instructed that not all items would remain just as cnarcy for the entire experiment – rather, on some blocks, certain items may (or may not) change in terms of how cnarcy they are, requiring participants to remain attentive to the feedback they received.

Specifically, the first part after the dash (from "rather" to "they are") is presumably something the participants are directly told (this is signaled by the change in tense). I think this reads strangely in a sentence that generally describes how participants are instructed.

Reviewer #3 (Remarks to the Author):

We greatly appreciate the thorough responses to our (and also the other reviewers') comments and concerns, and the revisions the authors have committed to the manuscript - we are thus happy to recommend the paper for publication.

Reviewer #4 (Remarks to the Author):

I co-reviewed this manuscript with one of the reviewers who provided the listed reports. This is part of the Communications Psychology initiative to facilitate training in peer review and to provide appropriate recognition for Early Career Researchers who co-review manuscripts.

Version 2:

Decision Letter:

Dear Mr Graham,

Your manuscript titled "Asymmetric learning and adaptability to changes in relational structure during transitive inference" has now been seen by our reviewers, whose comments appear below. In light of their advice I am delighted to say that we are happy, in principle, to publish a suitably revised version in Communications Psychology.

We therefore invite you to revise your paper one last time to address the remaining concerns of our reviewers and a list of editorial requests. At the same time we ask that you edit your manuscript to comply with our format requirements and to maximise the accessibility and therefore the impact of your work.

EDITORIAL REQUESTS:

SUBMISSION INFORMATION:

OPEN ACCESS:

*** TRANSPARENT PEER REVIEW:** Communications Psychology uses a transparent peer review system. On author request, confidential information and data can be removed from the published reviewer reports and rebuttal letters prior to publication. If you are concerned about the release of confidential data, please let us know specifically what information you would like to have removed. Please note that we cannot incorporate redactions for any other reasons.

*** CODE AVAILABILITY:** All Communications Psychology manuscripts must include a section titled "Code Availability" at the end of the methods section. We require that the custom analysis code supporting your conclusions is made available in a publicly accessible repository at this stage; please choose a repository that generates a digital object identifier (DOI) for the code; the link to the repository and the DOI must be included in the Code Availability statement. Publication as Supplementary Information will not suffice.

*** DATA AVAILABILITY:**

Link Redacted

Best regards,

Troby Lui

Troby Lui, PhD
Associate Editor
Communications Psychology

REVIEWERS' COMMENTS:

Reviewer #2 (Remarks to the Author):

The reviewers adequately addressed all my remaining concerns, and I am happy to recommend acceptance.

‘Asymmetric learning and adaptability to changes in relational structure during transitive inference’ - Reply to Reviewers

REVIEWER EXPERTISE:

Reviewer #1: transitive inference, reinforcement learning

Reviewer #2: decision making, reinforcement learning

Reviewer #3: transitive inference, reinforcement learning

Reviewer #1

Summary

This paper examines how people adapt to changes in relational organization through incremental learning from local comparisons. The authors use a transitive inference (TI) task in which participants first learn a relational hierarchy of 7 items, but halfway through the task there is a sudden change such that either the lowest- or highest-ranked item moves to the other end of the hierarchy. Following on earlier work showing that people demonstrate asymmetric belief updating in a similar type of task, here the authors examine whether such asymmetric biases in learn help or hinder the ability to adapt to a sudden change of this sort. Using relatively simple model-free reinforcement learning models, they show first through simulation that a winner-based learning policy of the kind seen in previous work would lead to successful learning of a positive change in rank (i.e., where the lowest-ranked item becomes the highest-ranked) but comparatively worse learning of a negative change in rank (i.e., where the highest-ranked item becomes the lowest-ranked). Results from the behavioral experiment show that 1) participants largely exhibit a winner bias in belief updating, replicating a previous study, and 2) after the change point, participants readily adapt to the upward change in rank, but show impaired learning after the downward change in rank. The RL modeling indicates that models with asymmetric value updating (Q-asymm) generally outperform symmetric value updating (Q-symm) in fitting these data. Further exploration of the results indicated that there were some participants who initially exhibited strong biases toward winner-based updating, yet who readily adapted to downward shifts in rank. The authors present a further model (Q-adapt) that adaptively changes the degree of bias depending on the strength of the preference between two items, where violations of strong predictions lead to symmetric updates, but asymmetric updating may still apply for comparisons with more uncertainty about relative value.

This paper has several notable strengths. It is generally well-organized and well-written, with a clear rationale and in-depth coverage of relevant past work. The methods and analyses appear to be appropriate for the research question and hypotheses. The authors have provided an open repository that includes the raw data, experiment files, and analysis files, and seem reasonably well-documented to support reproducibility.

The core behavioral findings are interesting and constitute a meaningful contribution to this research area. The authors replicate their previous finding of a winner-bias in belief updating in this task, helping to establish the robustness of that effect. The more novel finding is that, despite relatively minimal changes in feedback after the change point, people clearly seem less able to adapt to downward shifts in rank, including their knowledge of items whose ranks didn't change. The authors propose a reasonable mechanistic theory that might explain this phenomenon and which meaningfully builds upon the idea of an updating asymmetry.

There are a couple of weaknesses that stood out to me. The first concerns the large amount of data that was excluded (see point 1 below). The other broad concern is that, while the Q-adapt model is an interesting proposal, the evidence for it seems somewhat weak on the whole given the current data. The model recovery simulations don't imply "robust recoverability" to me, as there's a fair amount of confusion between the Q-asymm and Q-adapt models. The Q-adapt model also did not outperform the Q-asymm model on one of the criteria (comparisons of AIC scores). It's also unclear whether there's evidence for the trial-by-trial adaptation of learning rates proposed by the authors vs. a more discrete difference between the two phases (see point 4 below). In general, the Q-adapt model seemed to introduce a great deal of further complexity on top of the Q-asymm model, leaving it a bit unclear what aspects were essential for capturing the behavioral effect. That being said, while it doesn't seem all that parsimonious to me, I think the authors nicely explain the rationale for the proposed model and how it connects to related ideas in RL, and in that sense the theoretical contribution may outweigh the somewhat tentative nature of the evidence for the proposed mechanism.

We would like to thank the reviewer for their favourable evaluation of our work and for their thoughtful and helpful suggestions for further improvement.

Main points

1. Nearly half of the participants were excluded based on their performance prior to the change-point. I understand the reason for focusing only on people who demonstrate some learning of the hierarchy, but such a large proportion of excluded data is a bit surprising and may speak to the relative difficulty of the task (or perhaps broader concerns about data quality with an online sample). The ramifications of this for the generalizability of the conclusions are unclear to me (e.g., perhaps the observed winner asymmetry is overestimated because it was less common among the lower-performing participants who were excluded). I think the authors should minimally acknowledge the rate of exclusions more directly in the main text of the paper, and consider discussing the extent to which this may impact their conclusions. It may also be helpful to report other signals of data quality that could corroborate exclusions if they are available.

We appreciate this comment and now acknowledge more explicitly in the main text that the relatively high exclusion rate may pose a potential limitation of our study, while also including figures and key statistical analyses for the participant set obtained following more liberal inclusion criteria (see 'Limitations' section in the revised *Discussion*, p21, lines 576-586, and Fig S8 and S9A-B). Identifying participants who were capable of learning the transitive hierarchy to a sufficient degree, was a necessary step in order to probe any changepoint-related changes in learning. Nonetheless, to ascertain the robustness of the pre-changepoint winner-biased learning asymmetry to these exclusion criteria, we re-calculated the model-agnostic measures of learning asymmetry - i.e. the slope of the relationship between TI accuracy and combined pair value for all pre-changepoint trials -, but for the full set of 150 participants ($N_{\text{down}} = 74$; $N_{\text{up}} = 76$). Reassuringly, we nonetheless observed asymmetry slopes that were significantly below 0 in both groups, such that increases in combined pair value on TI trials were associated with a decline in accuracy ('up': mean $\beta = -0.02 \pm 0.005$ SE, $t(75) = -4.03$, $p < .001$, $d = -0.46$, 95% CI = (-0.03, -0.01); 'down': mean $\beta = -0.02 \pm 0.005$ SE, $t(73) = -4.73$, $p < .001$, $d = -0.55$, 95% CI = (-0.03, -0.01)). As in the post-exclusion participant set, this degree of asymmetry did not significantly differ between groups ($t(148) = 0.57$, $p = .570$, $d = 0.09$, 95% CI = (-0.01, 0.02)). Together, these additional analyses provide evidence that the value compression effect and key model comparisons were relatively insensitive to our performance-related exclusion criteria.

2. The authors highlight the intriguing result that a downward shift in rank seems to impair TI performance for other, non-anchor items (items 2-6) despite nothing changing about the feedback received for those items. I struggled a bit to understand how this arises in the model, and I think the authors could clarify the effects on "downstream inferential knowledge" (e.g., by moving the explanation in lines 174-178 to somewhere earlier in the paper). There are some early claims about these effects which are somewhat vague and make it difficult to appreciate that it's not only the slower downward adjustment of item 7, but also the fast upward adjustment of item 1 that together produce worse discriminability for lower-ranked items overall.

We agree, and now provide additional explanation early in the text (see p3 lines 79-82). Specifically we have included extra detail in the sports team example in the introduction in order to clarify how this downstream impact arises, as well as further clarification in the caption of Fig 1C. Thank you for this helpful suggestion.

3. The authors report a basic analysis indicating that participants adjusted equally well to the upward and downward shifts in terms of comparisons involving the anchor items (1 or 7). It wasn't clear to me why the change in the anchor items wasn't included in Figure 2 showing the changes over time, since the models also make different predictions about the speed with which estimates of those items change. Although the overall comparison wasn't significant, the means at least suggest a slower adjustment of item 7 for the downward shift compared to item 1 in the upward shift. In any case it seems an important feature of the data to provide alongside the changes in accuracy for non-anchor items (and which I found hard to grasp from the matrices in Figure 3).

The reviewer raises a thoughtful question, how these changes in anchor items may have differed between groups. The change in rank of these anchor items precluded us from including comparisons for which the underlying ground truth had reversed in our metrics for overall post-changepoint TI performance, owing to the confounding influence of participants' pre-changepoint accuracy on these trials. For example, comparisons involving i_1 were mostly of lower combined value, and thus tended to be more easily inferred by participants than (generally higher combined value) comparisons involving i_7 by virtue of the value compression effect outlined in the paper. This means that, if the pre-changepoint accuracy on trials involving i_1 were already high, then changing how accuracy is coded on subsequent trials involving this item in the 'up' condition would give the impression of a sudden and sharp decline in accuracy, given participants' hitherto strong preference for $i_1 < i_n$. We therefore restricted our analyses on comparisons involving non-anchor items to not only avoid this skewing effect of previous response preferences, but also, more importantly, to examine whether changing the ranking of each anchor item differentially impacted participants' downstream capacity for TI on trials involving intermediate items whose value had not changed.

Nonetheless, as the reviewer rightfully points out, the asymmetric learning framework predicts differences in how agents will update their beliefs about each of the anchor items. This presented us with the challenge of how to illustrate changes in anchor choices over time alongside accuracy for other TI pairs, as in Fig 2, since including such anchor items in Fig 2 would give the impression of an artificial sudden drop in accuracy at the changepoint. The more relevant metric for probing participants' learning of anchor items was there for the change in preference with respect to these items (i.e. the probability that these items were chosen), obviating the need to code trials using a switching ground truth. We have therefore included Fig S3, which illustrates the development of participants' preferences for the moved and unmoved anchors in each group.

In addition, we have included an additional passage analysing participants' preferences for the *unmoved* anchor item (i.e. i_7 for the 'up' group; i_1 for the 'down' group) (p11-12, lines 269-281). The two groups of participants exhibited differences in how their preferences for this unmoved anchor item changed as a result of the changepoint - whereas 'up' participants' preference for i_7 remained stable, 'down' participants increased their preference for i_1 . In other words, although the ranks of these unmoved anchors did not change, 'down' participants nonetheless exhibited a bias towards increasing their tendency to choose the unmoved anchor i_7 . We believe that this further motivates our adaptive asymmetry hypothesis, since it indicates that although participants were equally capable of changing their beliefs about the moved anchor, this more symmetric updating of the moved anchor existed alongside more winner-biased of the lower anchor in 'down' participants, even though its rank had not changed.

4. As I noted above, the Q-adapt model involves a big jump in complexity, which left me wondering how well it could be validated with the current data. The authors also fit a simpler model (described at bottom of pg. 13) with separate learning rates for each phase (pre- and post-changepoint), but this is not included in the model comparisons. This model could be seen as a simpler version of the "adaptive asymmetry" explanation: That participants adjust their learning rates once after receiving feedback that strongly violates existing beliefs. I'd be interested in how this compares to the other models, and more generally whether there's any other evidence for the trial-by-trial adjustment of learning rates based on the confidence for the current pair.

We appreciate the reviewer's concerns about the extent to which *Q-adapt* best captured participants' behaviour. We acknowledge that the model exhibited only a slight advantage in ppx over *Q-asymm*, with the comparison of AIC model evidence scores being even narrower. Given this relatively fine balance in the predictive performance of these two models, we therefore turned to their generative performance - i.e. how well these models, when simulated using the best-fitting parameters, were able to recapitulate the key behavioural effects observed in humans. In this respect, *Q-adapt* outperformed *Q-asymm*, with the most striking discrepancy lying in how these models reproduced the winner-biased learning asymmetry in pre-changepoint data - whereas participants' asymmetry indices were distributed bimodally under *Q-asymm*, indicating an even spread of winner- and loser-biased agents, the proportions of participants identified by *Q-adapt* as winner-biased aligned much more closely with those derived under our model agnostic asymmetry slope. Given *Q-adapt*'s ability to capture participants who exhibited pre-changepoint value compression, but who nonetheless were able to appropriately adapt to the 'down' changepoint, we interpreted this superior generative performance as 'tipping the scales' in favour of *Q-adapt* over *Q-asymm*.

Nonetheless, we are also mindful of the increase in mechanistic complexity that *Q-adapt* introduces, and that the ability of well-performing 'down' participants to adapt to the changepoint,

even despite exhibiting a winner-bias before the changepoint, could also be accounted for by a simpler model with separate learning rates for each phase - i.e. *Q-asymm*². In re-fitting this model to participant data and including it in our model comparisons, we observed that *Q-asymm*² more convincingly emerged as the winning model in both groups under AIC, and under p_{XP} derived from AIC, and have amended our Results section to reflect this (e.g. p14-15, lines 351-364). Incidentally, conducting model comparison using BIC revealed a more mixed picture, with *Q-adapt* and *Q-asymm*² emerging as the winning model in the 'up' and 'down' groups, respectively (which we note in p15, lines 364-367). However, our models exhibited much poorer recoverability under BIC. We therefore focus our analyses in the main text on AIC, while also including these BIC-derived comparisons in Fig S6A-B.

In observing that *Q-asymm*² emerges as the best-fitting model, we believe that this does not change the central findings of our study - i.e. that most participants exhibit an initially winner-biased learning asymmetry, which confers different degrees of sensitivity to each changepoint, but with the possibility that well-performing participants exhibit adaptability in their learning asymmetry). The key difference lies in the claims that are made about how this asymmetry arises. *Q-asymm*²'s 'one-shot' switch to its post-changepoint learning rates come into effect upon receipt of the first " $i_2 < i_1$ " feedback, but is agnostic about how this adaptability emerges, and hence how any modulation in learning asymmetry may be adapted to the demands of the task.

In contrast, in formalising a role for trial-by-trial choice preferences in shaping learning asymmetry, we presented *Q-adapt* in an attempt to unite our present findings with those of Ciranka et al. (i.e. the observation that humans tend to show my symmetric learning under full feedback regimes), while also drawing on empirical and theoretical work on the role of decision noise in shaping magnitude compression effects (Spitzer et al., 2017; Juechems et al., 2021).

These two models each have their own merits with respect to their predictive and generative performance. While convincingly outperforming other candidate models in terms of model evidence, *Q-asymm*² did not reproduce the significant changepoint x direction interaction effect on non-anchor TI accuracy. In contrast, the reverse is true for *Q-adapt*. In this respect, we believe that these models complement one another - one model provides a superior overall fit to participants' responses, while the other formalises a hypothesis about the basis of adaptive asymmetry in such a way that it reproduces the key behavioural effect of interest, albeit with a worse overall fit.

While this presents something of an open question as to which model ought to be advanced as the definitive winner, in either case, we believe that the model comparison corroborates our key claim that winner-biased learning asymmetries a) differentially advantage humans towards certain relational changes, while b) being malleable. In other words, learning asymmetry is adaptive in both cases - either in the trial-by-trial sense (*Q-adapt*), or in a more discrete sense (*Q-asymm*²). We have therefore amended our manuscript to better reflect this. More specifically, we have outlined the

merits of each of these two models and highlighted that *Q-asymm*² provides the best fit to participant data (e.g. p17, lines 438-443; p19, lines 514-516). In turn, we have softened our claims about how asymmetry adaptability may emerge while nonetheless making the reader aware of the potential theoretical proposal provided by *Q-adapt*, and highlighted the importance of future work in investigating the computational roots of such adaptability (p20, lines 528-539).

Reviewer #2

Summary

Graham and Spitzer report a study that aims to demonstrate that when people update relational representations, they adapt the degree to which they use an asymmetric learning possibility. They found that participants easily learned that a previously low-ranked item now had a higher rank. However, participants had a harder time learning that a previously high-ranked item now had a lower rank. Computational modeling suggested that participants exhibited a winner-biased learning policy, but adapted this strategy when they needed to learn a new relationship with a new inferior option.

This is an interesting paper, with a sophisticated computational modeling technique and convincing data. The paper is not squarely in my field of expertise (which is in decision making and computational modeling, but more on the side of cognitive control), so my comments may come a bit out of left field (which may be good given the broader readership of this journal). I don't have many comments about the design of the experiment, and the analysis of the data. These appear quite solid.

We appreciate the reviewer's positive assessment of our manuscript and their insightful suggestions to enhance the quality of the work.

Main points

My major concern, however, is that the scope of this paper appears relatively narrow. From my (outsider) understanding, it was already clear that people use a winner-biased updating rule. If this is the case, then it should be relatively obvious that people are going to perform better in the "up" compared to the "down" condition. The only remaining novelty then is that people seem to adapt their learning policy based on the context. I am worried that this may not be a big enough

advancement in understanding in the field to warrant publication, but I'll readily admit that I am not in the best position to comment on this.

We appreciate the reviewer's thoughtful feedback and their concern about the perceived novelty of our study. However, we would like to clarify why we believe our findings represent a meaningful contribution to the field of learning and decision-making.

While the positivity bias in learning has been well-documented in bandit tasks, its generalisability to other domains remains an active area of research (Palminteri & Lebreton, 2022). Recent work has begun to explore its relevance in foraging contexts (Garrett & Daw, 2020), but its role in inference and structure learning has only recently been addressed (Ciranka et al., 2022). Investigating the kinds of learning biases exhibited by humans during such structure learning tasks is an interesting extension in itself, because it has implications for how we acquire relational knowledge, as opposed to updating beliefs about payout options. In parallel, while previous research has explained magnitude compression effects in terms of Weber scaling - i.e. compression at the representational level -, Ciranka et al. provide an alternative view: that such compression effects can arise as a product of the learning update itself, thereby tying together reinforcement learning biases with literature on more generalised cognitive distortion.

Critically, positing such learning asymmetries in the context of structure learning entails the – hitherto untested – novel prediction that humans ought to differ in their sensitivity to different changes in relational structure. Importantly, this should also manifest itself in how people's 'downstream' knowledge of unchanged relations is also impacted, which we test, for the first time, in the present study. We therefore believe that our study provides a key contribution to the cognitive science of learning and decision making by not only replicating the previously observed asymmetry during inferential learning, but also by verifying the resultant hypotheses regarding people's ability to adapt to changes in relational structure. In addition, we offer potential novel refinements of our computational modelling of inferential learning to encompass adaptive adjustments in dynamic/changing environments, which, to the best of our knowledge, have not been considered in earlier TI work.

I think Figure 2 can be improved. First, it is not immediately obvious what the different colors represent. There is a little matrix on the right that also contains these colors, and some labels ("low", "medium", "high") but one needs to read the caption to understand that this is the true difference in rank. More importantly, however, the moving window analysis induces apparent changes in behavior before the onset of the changepoint. For example, if we look at human behavior in anticipation of the change point, it looks like humans already start adjusting the value of the items (especially in the "down" condition where they seem to flatten before the changepoint in comparison to the "up" condition). I recommend that the authors find another way of plotting the data, so that post-changepoint adaptations do not bleed in the pre-changepoint behavior.

We acknowledge that participants' accuracy as a function of the changepoint and the pair value could have been made clearer in Fig 2, and so thank Reviewer 2 for pointing this out. We did experiment with adopting a sliding window approach that treats each phase of the experiment separately - i.e a smoothed curve for the pre-changepoint data, along with a separate smoothed curve for the post-changepoint data. However, restarting the binning procedure for these two separate phases introduced discontinuities between the end of the pre-changepoint curves and the start of the post-changepoint curves, since these correspond to truncated bins containing fewer trials that are therefore more noisy. At the same time, the reviewer is right about the bleeding of pre- and post-changepoint behaviour into one another. To address this, we have made two alterations to the plot: firstly, we now code each datapoint (i.e. each bin centre) using its relative trial number, i.e. when the trial was presented, relative to the first post-changepoint trial. Centring participants' choices around these changepoints ensures consistency in the binning procedure between participants, whose first encounter with the changepoint $i_1 < j_1$ may differ, owing to the randomisation of trial sequences. The red shaded section of Fig 2 therefore now aligns with this first changepoint trial for all participants (i.e. 'trial 0'). Secondly, we have reduced the bin size from 100 to 70 trials, to reduce the amount of leakage between changepoint phases.

Regarding the colour-coding of the low, medium and high-valued comparisons, we have added a title and axes labels to this choice matrix in order to highlight that this refers to the combined pair value of each comparison.

The paper never makes it clear what "cnarcy" means. I surmised that this is a made-up word, but the authors never explicitly state this. I think the purpose of using this word deserves a little bit more attention. I am also not quite sure why the authors chose this non-word, because (at list in my mind) it sounds like "snarky", which is a real English word. I understand that the study was conducted in Germany, but many young Germans are quite comfortable with English (especially in higher education), and so I am afraid that they may use this definition of the word to make their initial judgments. This would make their initial rank order not so neutral. Of course, this concern is present in both the up and down conditions, so its effect should cancel out in the long run. Still, this potential interpretation of the word stood out to me.

'Cnarcy' is indeed a made-up term, deliberately chosen to avoid any direct semantic associations that could influence participants' judgements. The rationale behind using this word was to ensure that participants inferred item rankings purely from the feedback they received, rather than relying on pre-existing knowledge or intuitive heuristics. We have added a sentence in the 'Stimuli, Task and Procedure' section of the Materials and Methods of the revised manuscript to make this more explicit (p23, lines 633-634).

While we understand the reviewer's concern that 'cnarcy' may sound somewhat similar to the English word 'snarky', we do not believe this would introduce systematic bias. 'Snarky' describes a tone or attitude, and is not easily applicable for ranking the inanimate objects used in our study (e.g. scarves, calculators, or telephones). Whereas words that directly imply a ranking dimension (e.g., 'expensive', 'good', 'strong') would have been problematic, 'cnarcy' does not impose any obvious ordering principle on the item set. Furthermore, not only were participants explicitly informed that cnarciness was unrelated to any characteristics that the objects had in real life, the item-rank assignments were randomised across participants, ensuring that any individual interpretation of 'cnarcy' could not systematically influence the results. Since participants had to infer the relative rankings from pairwise comparisons and feedback, we think that any potential preconceptions about the term – if they existed at all – would not persist beyond the early trials of the experiment.

It was not clear to me from the text how "adjacent and TI trials were randomly interleaved within each of six blocks". Can the authors add more detail?

We have included additional detail on how these trial sequences were pseudo-randomly interleaved in the 'Stimuli, Task and Procedure' section of the Materials and Methods of the revised manuscript (p23, lines 648-653). Specifically:

"Each possible stimulus pairing (N=21) was repeated with the left and right positions of items reversed. Each adjacent trial pairing (N=6) was additionally repeated twice more (i.e. once with each L/R configuration), giving a total of 54 trials per block, of which 24 provided feedback and 30 provided no feedback. Trial sequences were pseudo-randomly interleaved, splitting each block into two shuffled sub-blocks within which each of the 21 possible item pairings (along with each of the six repeated adjacent pairs) was observed before being flipped and presented again in the second sub-block."

When the authors inspect the development of TI accuracy after the changepoint, they divided this phase in half and used the "half" as a within-subjects factor. This was surprising to me, because trial number is a nice continuous measure that already represents an increase in time. Why did the authors make this predictor categorical? It seems like a lot of useful variance is thrown away by doing this. For example, they could just add trial number as a continuous regressor in an LME model.

We thank the reviewer for raising this suggestion about how to more clearly precisely examine time-dependent changes in inferential performance. In the revision, we have removed the categorical analysis (ANOVA) in the 'Differential Impact of Changepoint on TI Performance' section

of the results that divides the post-changepoint phase in half. Instead, we now report a logistic mixed-effects regression analysis with trial number as metric predictor, as suggested by the reviewer (p11, lines 242-255). This improved analysis replicates a significant interaction effect, which corroborates our claim that the changepoint had a more detrimental impact on participants' downstream inferential knowledge in the 'down' group relative to the 'up' group. Thank you for this helpful suggestion.

It seemed arbitrary to me that the learning rate for the *Q-adapt* model can't reverse. The authors cite some literature suggesting that this does not happen, but the task certainly invites a reversal (and it seems optimal as well). Why not fit a model that allows for reversals to happen, and investigate whether relieving this restriction provides a better model fit.

This is an astute point, which we believe is in parts addressed in our response to Reviewer 1's 4th main point. The reviewer is correct that our *Q-adapt* model does not allow for learning rate reversals, owing to how its asymmetry modulator λ - i.e. the degree to which learning rates are distributed (a)symmetrically - is a quadratic function of its choice preference. More specifically, when the choice preference is weakest - i.e. $p(x < y) = 0.5$ -, then λ approaches 1, resulting in maximal winner-biased asymmetry, whereas when the choice preference is strongest - i.e. $p(x < y) = 0$ or 1 -, then λ approaches 0, resulting in maximal symmetry. The symmetrical nature of this quadratic function, and the way it peaks at $p(x < y) = 0.5$, means that it is impossible, in *Q-adapt*'s current form, for there to be a choice preference strength that gives rise to a reversal in learning asymmetry. Rather, allowing the model to switch its learning asymmetry would require formulating an alternative model.

To this end, and in response to other reviewers' feedback, we re-fit participants' choices using *Q-asymm*². This model contains a separate pair of winner and loser learning rates for the pre- and post-changepoint phases, and so in that sense is able to reverse its learning asymmetry after the onset of the changepoint. This model provided a superior fit to participants' choices when considering its predictive performance - i.e. the strength of the model evidence relative to other candidate models -, but, unlike *Q-adapt*, was unable to show strong generative performance - i.e. the ability to generate the key behavioural effects of interest in simulations. In either case, however, we believe that the model comparisons support our central argument that winner-biased learning asymmetries result in differential sensitivity to relational changepoints, while also remaining adaptable. In advancing *Q-asymm*² as the winning model, with the caveat that *Q-adapt* displayed superior generative performance while also providing a theoretically grounded account of how its asymmetry adaptability emerges, we therefore now more explicitly indicate that well-performing participants in the 'down' group may alternatively have adapted to this changepoint by reversing their learning asymmetry. We thank the reviewer for these very thoughtful comments,

which encouraged us to provide a more balanced discussion of the modeling results in the revised manuscript (see also our response to Reviewer 1).

Reviewer #3

Graham and Spitzer present an interesting and clearly laid-out study that investigates the role of asymmetric reinforcement learning in learning generalizable relations; in particular, in the naturalistically relevant case where subjects must learn changes in underlying relational structure. Relational learning and the transitive inference task paradigm they adopt are important areas in the study of cognition, and the RL-based models they focus on are an important model class in clarifying learning mechanisms in humans and animals. This study is an extension of recent innovative work from their group (Ciranka et al. 2022) that showed that asymmetric learning (i.e. here, in the transitive inference task paradigm, updating of model parameters preferentially for items that are either “winners” or “losers”) actually facilitates relational learning, and that such learning well-matches experimental behavioral data. In our opinion, the themes of this work will be of interest to researchers across different fields (relational learning, RL). We also highlight that the authors’ aim to investigate cases of change to underlying relational structure represents a valuable extension to existing relational learning studies.

The authors extend their investigation of asymmetric learning more specifically to changes in relational structure by introducing a variant of the transitive inference task in which there is a shift in transitive (rank-based) structure, either where the highest- (or lowest-) ranking item becomes the lowest- (or highest-) ranking item. The authors collect data from human subjects on this task variant, from which they report a number of findings, mainly (1) human subjects generally perform better when a lowest-rank item is maximally promoted in rank (rather than when the highest-rank item is maximally demoted), in agreement with the basic expectation from their previous work, (2) across the task changepoint, a subset of human subjects appear to adapt their asymmetric learning from winner-based to loser-based, doing so in a manner correlated with task performance, and (3) an RL model that the authors introduce, which has the ability to adapt its asymmetric learning bias, provides a relatively better match (than a non-adaptive model) to the human data.

Overall, we found this study to be interesting and thorough, and we appreciate the clarity of the authors’ written presentation. We do however have several points that we would like to see addressed (main requests), in addition to some additional requests for improving the manuscript.

We are grateful to the reviewer for their encouraging and extensive feedback, offering valuable recommendations that have helped improve the paper.

Main points

- We request some clarification regarding the explicit prompting of the subjects to expect a change in relational structure (in lines 678-681 in Methods, the authors write: “At the start of the experiment and before each block, participants were reminded that not all items would stay as cnarcy for the entire experiment. Rather, on some blocks, certain items may or (may not) become more or less cnarcy, meaning their relations to other items (as reflected in choice feedback) may change.”) While we have no problem in principle with this aspect of the task, we believe a few things are appropriate:

- First, we think the authors should clarify in the manuscript (ideally in both main text and Methods) what the subjects were informed of regarding the possible changes to ‘cnarciness’, i.e. whether it would change at any trial (or for an entire block) and whether the change was completely arbitrary (i.e. a single item or any set of items? any kind of specific change in rank?). It would probably be most helpful simply to have the actual prompt used directly reported to the Methods section (and also with some adequate clarification in the main text).
- Second, the authors should provide some additional discussion (either in the Methods or, even better, in the Discussion) of how this point might alter the asymmetry adaptability or other learning/decision aspects (e.g. winner-/loser- asymmetry bias, decision noise) of the human subjects. At the very least, it would be helpful to clarify where the authors stand regarding the expected generality of their results and some insight into their rationale.
- We think these are reasonable requests since in many (if not most) real-world scenarios (even the authors’ example of teams winning and losing) there are clearly situations where there is not an explicitly prompted expectation of relational change. Further, it is reasonable to expect that such prompting could promote asymmetry-based adaptability. We do not think such a possible effect makes the findings less interesting, but would better delineate the impact of the findings in the paper if addressed openly.

Thank you for these comments - we are glad to provide further clarification. We have carefully revised the main text (p5, lines 123-125), and included more detail in the *Materials and Methods* section (p23, lines 656-660) to better reflect the exact wording of the instructions. More specifically, we have clarified how participants were explicitly informed in the instructions section that on some blocks, certain items may (or may not) change in terms of how cnarcy they are. This would change their cnarciness relations to the other items in the hierarchy, and hence participants should remain attentive to any changes in feedback. They were also reminded of this in the inter-block pause, and hence to remain attentive to the feedback they received. No other cues were given about a) which items would change, b) how many items would change, nor c) on which blocks

they would change. The exact instructions delivered can also be found in the experiment repo in the folder 'expt/resources/quick_instructions'.

The point about whether one's expectation (or lack thereof) of a relational change may influence one's adaptability is interesting. Different real-world situations may afford different expectations about how a relational hierarchy may change. Indeed, changes in observations may be reflective of different features of the environment - that is, variability in feedback may be due to an objective change in the environment (i.e. 'volatility'), or simply due to irreducible noise in observations (i.e. 'stochasticity'). We believe that the sports teams case is a representative example of these different kinds of changes - e.g. a team could suffer a minor blip in form, resulting in a local change in the latent hierarchy of teams, or could plummet in this hierarchy because of an injury to their star player.

Many experiments more explicitly dissociate volatility from stochasticity in this way (e.g. Behrens et al., 2007; Piray & Daw, 2024). In contrast, our feedback was deterministic, meaning that, under the assumption that a participant would have learned the transitive hierarchy to a reasonable degree of proficiency by the time the changepoint appears in block 4, it should be relatively easy to notice the feedback that violates this initial ordering, and hence that a changepoint has occurred. It is worth noting that Ciranka et al. (2021) did include a version of the TI experiment in which participants received stochastic binary feedback with 80% validity, and nonetheless observed the same winner-biased learning asymmetry as we did (in the pre-changepoint phase), reinforcing the generalisability of the winner-biased asymmetry beyond deterministic feedback regimes. However, it remains an interesting open question how these belief-updating asymmetries may in turn interact with humans' ability to disentangle objective relational changes from stochasticity in feedback. While we already alluded to this separation between volatility and stochasticity in the *Discussion*, we have amended this section to more explicitly draw attention to how these different types of uncertainty may interface with humans' adaptability to different kinds of changes in the environment (p20, lines 530-539). Thank you very much for encouraging these improvements.

- The authors find an interesting behavioral pattern in the human subjects in the "down" group, which remains unaccounted for by their RL models, namely the decrement in performance in the high trials after the changepoint (seen in Figs 2 and 3). This seems to be quite a striking (interesting) experimental result, but apart from noting its existence, it is not discussed further by the authors. It seems some discussion is merited, perhaps ideally in the Discussion section, with regard to why the human subjects show this, and/or possible modifications to the Q_adapt model (or other model).

The reviewer correctly points out that, descriptively, there seems to be some discrepancy between observed and simulated accuracy on post-changepoint TI performance as a function of pair value, particularly in the 'down' group. For example, whereas humans show a particular decline in

performance on higher-valued pairs, this isn't recapitulated in the simulated models. Of note, our primary motivation for distinguishing between 'low', 'medium' and 'high' valued comparisons was to make the value compression effects more interpretable. We therefore clustered together different parts of the choice matrix in the Fig 2 inset plot with similar combined values (as opposed to, for example, plotting each combined pair value separately). Nonetheless, owing to the fact that the TI comparisons involving anchor items were excluded from this analysis, this parcellation of the choice matrix into three groups reduces the number of pairs assigned to each group. Given this reduced signal-to-noise ratio, we therefore treated these groupings as more for illustrative purposes, rather than the basis for any quantitative analysis of compression effects (which we instead implement via the combined pair value x accuracy regression slope).

However, we do offer some thoughts on how this discrepancy may arise as a result of some additional 'Weber scaling' of higher magnitudes in the 'down' group, which serves to further compress higher-valued items. We have included this in the revised *Discussion* section (p19, lines 490-494).

- It also seems appropriate to include some discussion (again, perhaps solely in Methods, but certainly possibly in the Discussion) regarding whether and how the use of explicit TI (where the transitivity- or rank-based relation is explicitly told to the human subject, "cnarciness") would have an impact on symmetric vs. asymmetric learning biases, and on adaptability as well (this interacts with our request above regarding the explicit prompting of possible relational change). Given that - both- the transitive relation and the possible reconfiguration of the transitive hierarchy are explicitly prompted in the task the authors use, it seems particularly appropriate to speak to this issue directly. Further, as work on TI in animals is basically implicit, providing some discussion would help guide thinking on possible connections to the wider literature on TI. For instance, though implicit TI tasks (i.e. where the subjects are merely tasked with choosing between stimuli, and are not prompted about ordering (e.g. "cnarciness")) are almost certainly more difficult overall, it seems plausible to us that subjects who do perform implicit TI may show asymmetric learning (and, when controlling for performance, perhaps even more asymmetric learning than in explicit TI). (Given that TI in human subjects has been found to occur successfully even when subjects are not consciously aware (e.g. Greene Spellman et al. 2001), something like winner-/loser- bias and adaptability seem plausible: perhaps subjects do not need to be consciously aware or semantically know about the task in order for them to exhibit asymmetric learning/adaptability). If this scenario or some other is reasonable to postulate, then we think it would be beneficial to various readers to have the authors weigh in.

We agree that different mechanisms for inference imply differing degrees of explicitness/implicitness. We note that recent work has sought to understand how different representational schemes, as induced by different learning regimes, may have an important bearing

on the process by which transitive relations are inferred (Berens & Bird, 2022; Zhou et al., 2023). Relatedly, in our *Discussion*, we also allude to how acknowledge that our model-free framework leaves open the possibility that TI learning could also be understood as a model-based process (p18, lines 467-481). Below, we provide some more context on the instructions delivered to our participants, as we believe our task setup does not preclude participants from performing implicit TI.

Our instructions informed participants that their task was to “learn how [the items in the experiment] relate to each other in terms of how cnarcy they are”, where cnarciness is a new kind of relation between items that they, as scientists, are tasked with investigating. In that sense, they were explicitly cued that they are engaged in a relational learning experiment in which the concepts of ‘more/less than’ determine the correctness of their choices. While these basic relations imply a (transitive) hierarchy, we did not, however, explicitly inform participants that such a hierarchy, or ‘ranking’ exists. Rather, we simply told participants that they could only learn about the cnarciness of items through choice feedback, which would itself only be delivered on some trials, but not others. We also did not tell people that feedback would only be given on comparisons of adjacently ranked items.

Given the lack of any overt reference to rankings, hierarchies or any notion of adjacency, we believe that our explicit instructions that participants must learn about cnarciness - and that there may be a change in cnarciness - do not necessarily imply that participants are not engaging in implicit learning of a item values. Indeed, the symbolic distance effect in both accuracy and reaction times - i.e. faster and more accurate responses on trials with a larger rank distance between comparanda (p8, lines 189-194) - lends support to the hypothesis that this inference process was more implicit than explicit. In contrast, TI via a more explicit, successive reactivation of paired associations - e.g. ‘ $A < B$, $B < C$, therefore $A < B$ ’ would predict an inverse symbolic distance effect on accuracy and speed (e.g. as in Kumaran & McClelland’s (2012) REMERGE model). We believe this supports the view that participants automatically inferred and represented the position of the items in the set, making comparisons based on these inferred values and thus performing ‘implicit’ TI.

Additional requests

- In the abstract / introduction, the term “choice preference strength” is used — however, we think that for first-time readers, this term/concept will not at all be clear. To address this, can the authors unpack the phrase or use an alternative phrase (it is fine also to retain this phrase, but in a parenthetical).

We define choice preference strength in terms of the choice probability with respect to “the

probability of choosing one item over the other (p14, lines 339-340). Note that in the revised manuscript, we no longer use this (potentially confusing) term in the abstract/introduction, but only by the time Q-adapt is properly outlined in the '*Adaptive Asymmetry*' section of the *Results*, where the notion of "choice preference strength" is elucidated more clearly.

- line 602-607: regarding positive or negative RPE — we suggest providing some further brief clarification of relationship (either here or earlier in manuscript) between RPE and winner/loser-bias. Currently the authors simply say "plausible accounts", but this is too vague to be useful.

Thank you for this suggestion. The sentences that follow this mention of "plausible accounts for how updates may be asymmetrically scaled during RL" aim to provide some context on these existing proposals. We have added a connective at the start of the following sentence (p20, lines 560-561) to make the link between these two points clearer.

- While the connection is indirect, there is some interesting degree of conceptual similarity between the present study's choice-preference (or "belief") based modulation (of asymmetry learning) and the certainty-based updating mechanism for transitive knowledge assembly in the Nelli et al. 2023 study. Perhaps it is worth incorporating some discussion of this parallel, and even possible neural-level basis or substrates (if any), in the Discussion section.

We thank the reviewer for highlighting this intriguing connection between the potential role played by certainty in asymmetry modulation in our study, and in shaping the more overt reorganisation of structural knowledge in Nelli et al.'s (2023) study. We have included a more explicit reference to Nelli et al.'s certainty-based updating in the *Discussion* (p20, lines 520-522).

- We think it may be appropriate to provide (in a supplement) informal presentation of some of the debriefing responses, if they shed any light on the asymmetry bias and/or on the adaptive asymmetry modulation. Of particular interest could be subjects that show the extreme shift from winner- to loser-based asymmetry according to the Q_asymm2 model (Fig. 4D). Though informal, this could help provide some intuition (affirming/disaffirming) whether and to what extent the learning biases were consciously accessible and deliberately executed.

We have added a short summary of some of the debriefing responses in the *Methods* section (p24, lines 678-685). While we were unable to glean any insights from these responses that might serve to support or disconfirm our central claims about asymmetry and adaptability, we did find it interesting that the participants showed large variability in how difficult they found the task. This

could possibly be followed up by future work focusing on interindividual differences, which would likely require larger participant samples).

- In the authors' previous study on asymmetric learning (Ciranka et al. 2022), they find that the RL model best matching experimental data (the model is termed "Q2* + P") incorporated some degree of direct pair learning. If only in the Methods section, can the authors provide some clarification why this model was not adopted in the present study?

We thank the reviewer for highlighting this difference between our model space and the full model space explored by Ciranka et al.. We have addressed this more explicitly in the *Materials and Methods* section (p25-26, lines 729-733), and have also noted in the *'Limitations'* section of the *Discussion* that the interaction between episodic processes (i.e. 'pair learning') and asymmetric RL may provide interesting avenues for future research (p22, lines 598-606). The goal of our present study was to isolate the role of asymmetry in generalisation and inference - or, more specifically, how asymmetry-conferred adaptability may impact downstream inferential knowledge - , rather than optimise fit to directly reinforced pairs. Indeed, the episodic component of Ciranka et al.'s winning '+P' model does not directly contribute to inference, but instead solely to direct 'pair learning' of adjacent relations. Thus, we did not include this additional model feature from Ciranka et al.'s paper in our model comparisons for reasons of parsimony.

- For Fig. 3., it appears to be appropriate to provide results from Q_asymm in this figure (in between Human and Q_adapt), if simply to provide consistency with other figures showing both Q_asymm and Q_adapt, unless there is a clear reason not to.

We have updated Fig 3 to replace *Q-adapt* with *Q-asymm*². The reviewer is right to point out that a comparison with the other candidate models may be useful here. However, owing to the number of choice matrices plotted per model in Fig 3, this would not all fit on the page in the main text. We have therefore included these extra plots for the other candidate models as supplementary figures (Fig S3A-B).

- For Fig. 3B, the authors' choice of a monotonic color scheme is inappropriate. To clarify differences in sign (and not just magnitude), they should use a diverging color scheme (such as red-white-blue, where red is negative values and 0 is white, and blue is positive values; or some other similar color scheme).

We have altered the colour scheme in the manner suggested. In addition, we have included text annotations to the choice matrices in Fig 3B to improve the legibility of preference changes around the zero-point, and also those that exceed the minimum and maximum of the colourbar anchors (which we narrowed in order to increase discriminability of the intermediate comparisons), and hence whose values are difficult to discern from their colours.

- Line 300: There should be a reference to Fig. 4C.

Thank you, we have corrected this oversight.

- Fig. 4D: Larger binsizes (perhaps twice as large) seem appropriate.

We agree that larger bin sizes are more appropriate here, and have updated the figure accordingly.

- Lines 327-339 (and Fig. S4). Perhaps a Spearman correlation is more appropriate than a Pearson correlation (or complementary). Further, it would be helpful to include the include correlation statistics in Fig. S4 caption instead of just in main text.

We acknowledge that Spearman's correlation may be appropriate here, since the difference in A_{post} and A_{pre} under $Q\text{-}asym^2$ - i.e. the x-axis of Fig 6A-B - is more bimodally distributed in the 'down' group (as in Fig 4D, left panel). This statistical test gives the same pattern of significant and non-significant results as using Pearson's correlation, but for conciseness, we have now opted to use Spearman's correlation. Since this plot has now been moved from the supplementary material into the main text, we believe that it is now clearer to the reader that these correlations are significant in the 'down' group, but not the 'up' group, by the time they have been referred to this plot in the main text.

- Lines 369-377: The introduction of lambda is a bit confusing and unintuitive on first read. To address this, we recommend two changes: first, by changing the parenthetical "(indicating a weak preference)" to "(corresponding a weak choice preference, or weak "belief")", and similarly for "indicating a stronger preference"; second, by changing "reflects" (line 370) to "is dependent on".

We have implemented these suggested choices in wording (albeit opting for "is a function of" rather than "is dependent on"). Note that in the revised manuscript, this section has been moved to the supplementary text (*Supplementary Note 1*) in the course of other revisions.

- line 385: “strong prior belief”. Here and elsewhere, is the word “belief” strictly warranted as equivalent to the strength of choice preference? (Even though likely true in many subjects doing the task.) Perhaps the use of scare quotes for the word “belief”, and/or perhaps the use of a phrase or description closer to “choice preference strength” is safer.

Given that our RL models rely on point-estimates of the value of each item, their behaviour is in turn guided by point-estimates of the probability of outcomes (i.e. the probability that $i_x < i_y$), rather than probability distributions over outcomes. This introduces some difficulty in connecting the notion of choice preference, which naturally follows from point estimates of probability, with the notion of belief or uncertainty, which typically is more appropriate in the distributional/Bayesian case. Nevertheless, we chose to use the notion of ‘belief’ in certain parts of the text in order to elucidate how the asymmetry variable λ draws on the information theoretic notion of choice entropy or the surprise of an outcome. In other words, there is greater asymmetry when the prior belief is weak (i.e. choice preference approaches 0.5), since the resultant choice feedback carries the most information about how this preference ought to be updated.

To increase consistency in how these concepts are used, we have amended the manuscript throughout in order to emphasise that it is the choice preference strength that determines the degree of asymmetric updating. Nonetheless, at points where we outline the theoretical motivation for the model, we still make a reference to the role of ‘prior beliefs’ as we believe this allows the rationale for the model, and its connections to the entropy function, to be more clearly communicated.

- line 307: “leading to inflated estimates of loser learning rates.” This is somewhat confusing, as it seems like the model could also lead to deflated estimates of loser learning rates. Perhaps there is a better way to state this.

Thank you for highlighting this. We have altered the wording in order to more clearly motivate the subsequent ‘*Adaptive Asymmetry*’ section of the *Results* - i.e. emphasising how it may be inappropriate to fit a single pair of learning rates to the whole dataset, therefore raising the possibility that such asymmetry may be adaptive.

- p. 404: “choice parameterizations” : It would be helpful for the authors to clarify what this phrase refers to, using a parenthetical.

We have included an example of what this refers to in the Supplementary Note 1, (“... e.g. via a low decision noise parameter.”)

- Fig. 5B: The caption should clarify what p_{xp} estimated frequency is.

We have amended the caption of Fig 5A-B accordingly - i.e. “the expected proportion of participants best described by each model”.

- p. 439-446 - In Fig. 3B, the authors should provide plots of change in TI accuracy instead of pre-post change in choice preference (or do so in addition to pre-post change in choice preference in an added subpanel). Not only because this is more directly what the main text is referring to (in line 435), and also because this would serve to clarify the change in behavior across the changepoint.

For Fig 3B, we aimed to illustrate how participants' tendency to choose one item over another changed as a function of the changepoint. We decided to focus on change in preference - i.e. the difference in $P(i_x < i_y)$, owing to the change in ground-truth induced by the changepoint, which complicated any attempt to calculate changes in accuracy with respect to the moved items (i.e. i_1 in the ‘up’ group, and i_7 in the ‘down’ group). In contrast, plotting the change in preference provides a ‘cleaner’ illustration of how participants’ behaviour with respect to these anchor items varied from one phase to the next.

Nonetheless, we recognise that this introduces some confusion in the context of Fig 3A, which plots the accuracy in each changepoint phase - i.e. the probability of being correct, where correctness for comparisons involving the moved anchor item is coded differently for the pre- and post-changepoint phases. To increase the consistency of these two subplots and therefore avoid this switching in how these choice probabilities are to be understood, we amended Fig 3A so that it also, like Fig 3B, refers to choice preferences, as opposed to accuracy. Thank you for encouraging these improvements for clarity.

- line 517: Instead of “once the changepoint was reached”, we would recommend “by the time the changepoint was reached.”

This more detailed description of *Q-adapt* has been moved to *Supplementary Note 1*, where we have amended the wording accordingly.

- line 558: “relative balance...” — it’s a bit odd to talk about “narrowing” a “balance” (these words don’t naturally go together) — perhaps consider “balance of positive and negative learning rates

may dynamically become more even” OR “difference between positive and negative rates may dynamically be eliminated, rather than change in sign”

The phrase has been removed in the course of other revisions and is no longer contained in the manuscript.

- line 602: consider adding “...behavioural adaptability in a relational setting” or “behavioural adaptability in the setting of a relational task”

We have clarified that this interest in the role of asymmetric adaptability in clinical populations could be focused on looking either at changepoint adaptability more generally, or relational adaptability more specifically.

- Methods (638-651): can you provide some direct reportage of overall performance in included groups at both criterion levels (0.01 and 0.1)?

We have now more explicitly drawn attention to this more liberal participant sample in the ‘Limitations’ section of our revised *Discussion* (p21, lines 576-586), and included figures and the results of our key statistical tests - i.e. the significant direction x changepoint interaction, and the superior model fit for $Q\text{-}asym^2$ relative to other models - when applied to this alternative sample set in Fig S8 and Fig S9A-B.

- line 839 — “Fig S4C” is cited but there is no included S4C panel in the supplementary file. If the authors meant to include some (at least examples) of model parameters that may be traded-off but were not, that would be helpful.

Apologies for the oversight - this was supposed to refer to (what has now become) Fig S10A-B.

'Asymmetric learning and adaptability to changes in relational structure during transitive inference' - Reply to Reviewers

Reviewer #2 (Remarks to the Author):

In this revision, Graham and Spitzer have thoroughly responded to my comments. I appreciate the new continuous mixed effects model, as well as the new Q-assym² model that allows for the order between positive and negative learning rates to be different pre- and post-change point. I think the paper is strong, and I am looking forward to recommending acceptance when the following remaining concerns are addressed.

It's good to see that this model provides a better fit to the data than Q-adapt. However, I am not particularly convinced by the set of simulations in which the authors show that the Q-adapt, but not Q-assym², predicts the change point * direction interaction observed in human behavior (where performance post change point increases for the 'up' condition but not for the 'down' condition). At the core of this criticism lies the observation that the simulations from both models seem to show this effect (see Figure 2 rightmost panels). Moreover, the stats of the sets of ANOVAs (effect sizes and p-values) are close, which makes me wonder if the two models truly make such different predictions. First, you want to show that the pre-post by interaction effect in the q-adapt data is stronger than in the q-assym² data (see Nieuwenhuis et al. 2011 for a formal description of this problem). Most importantly, however, it seems odd to base this conclusion on just one set of simulations. I understand the authors simulated one dataset for each model so they could perform the same set of analyses as on the behavioral data. However, the noise from these simulations makes it hard to determine whether the authors have truly discovered that the fits from q-assym² do not make this prediction, or whether they just found a dataset in which the interaction did not occur by random chance. (This concern is exacerbated by the relatively similar stats of the two ANOVAs.) A more convincing version of this analysis would be to run a large number of simulations (say 100 or 1000 sims), and then count how many times each model includes a significant interaction effect. That would rule out any uncertainty about random chance influencing this validation check of the data.

We thank the reviewer for highlighting the need to formally test whether the key change point x direction interaction effect on TI accuracy differed between models. In line with this suggestion, we included 'model' as an additional within-subjects factor (Q-adapt vs Q-assym²) to the mixed ANOVA reported in the main text, thereby allowing us to test for a three-way model x change point x direction interaction. This interaction effect was non-significant, indicating that the extent to which the simulated behaviour exhibited the original change point x direction interaction effect did not significantly differ

across our two winning models. We have updated the manuscript (lines 427-248; 444-450; 542-546) to clarify that, although *Q-adapt* appeared to produce a significant changepoint \times direction interaction and *Q-asymm*² did not, the magnitude of this effect did not differ significantly between models. Thus, the evidence for *Q-adapt*'s superior generative performance is suggestive but not statistically robust. This 'tips the balance' further in the direction of *Q-asymm*², owing to its otherwise superior predictive performance. In the interests of transparency, we nevertheless note that *Q-asymm*²'s simulated behaviour, all else being equal, does not perfectly align with empirical behaviour, while still emphasising that, regardless of this tension between predictive and generative performance, the central conclusion that learning asymmetries do not remain fixed over time remains well-supported.

Regarding the reviewer's concern about the number of simulations used in this posterior predictive check, it may be helpful for us to clarify our method for analysing our simulated models. It is true that we simulated one dataset per fitted model – i.e. the trial sequence experienced by the participant to which the model was fitted. However, we then performed any resulting statistical tests as part of our model validation step on the model's *choice probabilities*, obtained via a logistic choice function (Eq. 5), as opposed to binary choices sampled from these choice probabilities. In other words, there is no stochasticity in the simulation procedure – the output of these simulated models was a deterministic function of the best-fitting parameters and the trial inputs, meaning the procedure yields stable predictions, without introducing sampling noise, and hence the same ANOVA results each time. This approach is in line with best-practice in cognitive modelling (e.g. Wilson & Collins, 2019; Palminteri et al., 2017), as it avoids variability introduced by sampling and directly reflects the model's expectations about choice tendencies under the recovered parameters. We have amended the manuscript in the Results (line 405) and Materials and Methods (lines 815-816) to emphasise that all validation steps were conducted on choice probabilities rather than sampled choices, and hope that this addresses any lack of clarity in our original description.

I think it would also be good to mention that 'cnarciness' was a deliberately chosen nonword in the main text (where it is first encountered), and not just in the methods.

We have more explicitly stated in the main text (lines 114-116) that 'cnarcy' was a nonsense word with no semantic associations with the item set used in the experiment.

Finally, I found new sentence in the results section a bit awkward:

Participants were instructed that not all items would remain just as cnarcy for the entire experiment – rather, on some blocks, certain items may (or may not) change in terms of how cnarcy they are, requiring participants to remain attentive to the feedback they received.

Specifically, the first part after the dash (from "rather" to "they are") is presumably something the participants are directly_told_ (this is signaled by the change in tense). I think this reads strangely in a sentence that generally describes how participants are instructed.

We have tweaked the wording of line 126 to make it clearer that both aspects of the sentence refer to instructions that were given to the participants – that is, participants were told that not all items would remain as *cnarcy* for the entire experiment, *and they were also told that* items may or may not change on certain blocks. We hope that this improves the readability of this sentence.

Reviewer #3 (Remarks to the Author):

We greatly appreciate the thorough responses to our (and also the other reviewers') comments and concerns, and the revisions the authors have committed to the manuscript - we are thus happy to recommend the paper for publication.

Reviewer #4 (Remarks to the Author):

I co-reviewed this manuscript with one of the reviewers who provided the listed reports. This is part of the Communications Psychology initiative to facilitate training in peer review and to provide appropriate recognition for Early Career Researchers who co-review manuscripts.